



# Particulate organic matter controls benthic microbial N retention and N removal in contrasting estuaries of the Baltic Sea

Ines Bartl[1*], Dana Hellemann[2*], Christophe Rabouille[3], Kirstin Schulz[4], Petra Tallberg[2], Susanna Hietanen[2], Maren Voss[1]

[1]Department of Biological Oceanography, Leibniz-Institute for Baltic Sea Research Warnemünde, Seestr. 15, D-18119 Rostock, Germany

[2]Ecosystems and Environment Research Programme, University of Helsinki, 00014 Helsinki, Finland

[3]Laboratoire des Sciences du Climat et de l'Environnement, UMR CEA-CNRS-UVSQ and IPSL, Av. de la Terrasse, 91198 Gif sur Yvette, France

[4]Department of Estuarine and Delta Systems, Netherlands Institute for Sea Research and Utrecht University, P.O. Box 140, 4400 AC Yerseke, the Netherlands

[*]shared first-authorship

*Correspondence to*: Ines Bartl (ines.bartl@io-warnemuende.de)

## Abstract

Estuaries worldwide are known to act as "filters" of land-derived N loads, yet their variable environmental settings can affect microbial nitrogen (N) retention and removal and thus the coastal filter function. We investigated microbial N-retention (nitrification, ammonium assimilation) and N-removal (denitrification, anammox) in the aphotic benthic systems (here defined as: bottom boundary layer [BBL] and sediment) of two Baltic Sea estuaries that differ in riverine N loads, trophic state, bottom topography, and sediment type. Contrary to our expectations, nitrification rates ($5 - 227$ nmol $L^{-1}$ $d^{-1}$) in the BBL neither differed between the eutrophied Vistula estuary and the oligotrophic Öre estuary, nor between seasons. Ammonium assimilation rates were slightly higher in the oligotrophic Öre estuary in spring but did not differ between estuaries in summer ($9 - 704$ nmol $L^{-1}$ $d^{-1}$). In the sediment, no anammox was found in either estuary and denitrification rates were higher in the eutrophied ($349 \pm 117$ µmol N $m^{-2}$ $d^{-1}$) than in the oligotrophic estuary ($138 \pm 47$ µmol N $m^{-2}$ $d^{-1}$). Irrespective of their differences, in both estuaries the quality of the mainly phytoplankton-derived particulate organic matter (POM) - evaluated by means of C:N and POC:Chl.*a* ratios - seemed to control N-cycling processes through the availability of particulate organic N and C as substrate sources. Our data suggest, that in stratified estuaries, phytoplankton-derived POM is an essential link between riverine N loads and benthic N cycling and may function as a temporary N reservoir via long particle residence time



or coastal parallel transport. Even at low process rates, effective coastal filtering would thus be achieved by the increased time available for the recycling of N via microbial retention processes until its permanent removal via denitrification.

## 1 Introduction

Human nitrogen (N) utilization, especially in agriculture (Galloway and Cowling, 2002; Rabalais, 2002) has strongly increased riverine N inputs into coastal zones and therefore the eutrophication of coastal waters. The negative effects on coastal

ecosystems include oxygen deficiency and a loss of biodiversity (Diaz and Rosenberg, 2008; Rabalais, 2002; Richardson and Jørgensen, 2013). The coastal zone of the semi-enclosed Baltic Sea in northern Europe is particularly prone to eutrophication as it annually receives ~682 kt of waterborne total N (TN, average 1994–2014; HELCOM 2011, 2015, 2018) from a catchment area inhabited by > 85 million people (Sweitzer et al., 1996).

Estuaries are the primary recipients of these land-derived nutrient inputs and act as filters that reduce the delivery of riverine

N loads to the open sea. This reduction is facilitated by intense biogeochemical cycling (Nedwell et al., 1999; Soetaert et al., 2006) which can be categorized in retention or removal processes (Asmala et al., 2017). N retention is defined as the cycling of bioavailable N within a system for longer than its mean water residence time, and N removal as the permanent removal of N from a system via burial and the production of gaseous forms (Asmala et al., 2017). Microbial N transformation processes, such as uptake into biomass, ammonification, nitrification, and dissimilatory nitrate reduction to ammonium (DNRA),

contribute to N retention, whereas denitrification and anaerobic ammonium oxidation (anammox) lead to N removal. In coastal ecosystems, these processes are especially intense in the bottom region, which comprises the sediment and overlying bottom boundary layer (BBL), i.e., the turbulent water layer directly above the sediment (Richards, 1990). These two components of the bottom region, referred to herein as the benthic system, are closely linked via the exchange and diagenesis of solutes and particles at the sediment-water interface (Boudreau and Jørgensen, 2001).

Nitrification, the aerobic oxidation of ammonium ($NH_4^+$) via nitrite ($NO_2^-$) to nitrate ($NO_3^-$), and denitrification, the anaerobic reduction of $NO_3^-$ to nitrous oxide ($N_2O$) and di-nitrogen ($N_2$), are the key microbial processes of the coastal N-filter. Nitrification provides substrates for N retention (primary production, DNRA) and N removal (denitrification and anammox). It is mainly regulated by oxygen and the availability of $NH_4^+$ (Ward, 2008), which can lead to competition with ammonium assimilation. In coastal systems, particulate organic matter (POM) is an additional important controlling factor (Bartl et al.,

2018; Damashek et al., 2016; Hsiao et al., 2014), as nitrifiers are often found attached to particles (Dang and Chen, 2017), where they utilize the $NH_4^+$ generated during POM degradation (Klawonn et al., 2015). Denitrification is the dominating N removal process in coastal sediments (Dalsgaard et al., 2005) and is mainly controlled by concentrations of the substrates $NO_3^-$ and dissolved organic carbon (Piña-Ochoa and Álvarez-Cobelas, 2006). Equally important is the quantity (Deutsch et al., 2010; Jäntti et al., 2011) and quality of the POM, as the source of both N and C substrates (Bonaglia et al., 2017; Eyre et al., 2013;

Hietanen and Kuparinen, 2008).



The efficiency of N retention and N removal within an ecosystem is influenced by environmental settings. For instance, pronounced stratification of the water column in deeper coastal waters (>20 m) limits vertical mixing and separates riverine N loads from the benthic system and a long water residence time enables high N removal efficiency (Finlay et al., 2013; Nixon et al., 1996). In addition, in eutrophied ecosystems, the rates of nitrification and denitrification are often enhanced, due to the

higher availability of substrates (Damashek et al., 2016; Finlay et al., 2013). The sediment type influences the transport of substrates into, through, and out of the sediment and thereby impacts benthic nitrification and denitrification. In muddy, cohesive (non-permeable) sediments, diffusive and fauna-induced fluxes govern solute exchange, while in sandy, permeable sediments advective pore-water flow is an additional transport process (Huettel et al., 2003; Thibodeaux and Boyle, 1987). Advective pore water flow leads to an increased supply of oxygen, oxidized solutes, and particles into the sediment, as well as

the build-up of a complex redox zonation, that in sum increases organic matter turnover (Boudreau et al., 2001) and can favour high N turnover (Huettel et al., 1998, 2014).

Baltic estuaries are highly variable in terms of stratification, sediment type, and riverine N-load composition (Asmala et al., 2017; Conley et al., 2011). Nonetheless, the effects of this variability on N removal and N retention have rarely been studied. In a recent review, Asmala et al. (2017) estimated that the Baltic coastal zone removes ~16% of annual land-derived TN loads

via denitrification, while the remaining 84% is suggested to be retained within the coastal zone or exported to the open Baltic Sea. However, model results indicate that export of riverine N to the open Baltic Sea accounts for only a minor share of the TN load (Radtke et al., 2012). Although most N is probably retained within coastal zones, actual in situ process quantifications in coastal benthic systems of the Baltic Sea are scarce (e.g. Bonaglia et al., 2014; Jäntti et al., 2011).

Therefore, the aim of this study was to examine microbial N removal (denitrification, anammox) and N retention (nitrification,

ammonium assimilation) in the aphotic benthic systems of two Baltic estuaries with contrasting environmental settings. The small, northern Öre estuary receives low riverine N loads (430 t TN yr$^{-1}$, Table 1) from a catchment area of forest and bog (Wikner and Andersson, 2012). Its oligotrophic state is reflected in low concentrations of nutrients and total organic carbon and by its low primary production rates (Ask et al., 2016; Wikner and Andersson, 2012). In contrast, the 12-fold larger southern Vistula estuary receives high riverine N loads (97 000 t TN yr$^{-1}$, Table 1) from a catchment area of intensively cultivated

cropland (Pastuszak et al., 2012). High concentrations of nutrients and organic matter (Pastuszak et al., 2012) and high primary production rates (Wielgat-Rychert et al., 2013; Witek et al., 1999) have led to its eutrophied state. The estuaries further differ in their bottom topography and sediment composition, while both estuaries are non-tidal, stratified, and receive their peak riverine N-loads in spring. The different environmental settings were anticipated to result in different rates of microbial N retention and N removal and thus in different estuarine filter efficiencies. Previous studies indicated that POM is an important

factor controlling denitrification in the sediment of the Öre estuary (Hellemann et al., 2017) and nitrification in the BBL of the Vistula estuary (Bartl et al., 2018) suggesting that POM plays an important role in the overall coastal N-filter function, irrespective of the environmental setting.



To address these questions, published results of sediment characteristics and N removal in the sediment of the Öre estuary (Hellemann et al., 2017), and water column characteristics and N retention in the BBL of the Vistula estuary (Bartl et al., 2018)

are combined with unpublished environmental data and process rates from both estuaries.

## 2 Materials and Methods

### 2.1 Study areas and sampling

The Öre estuary is located on the Swedish coast of the Quark Strait, northern Baltic Sea (Fig. 1). It covers an area of ~71 km² and has a volume of ~1 km³ (SMHI, 2003). Inputs into the estuary originate from the Öre River, whose mean discharge is 36

m³ s⁻¹, creating a river plume of 2–3 m vertical and ~10 km horizontal extent (Forsgren and Jansson, 1992). The water turnover time (estuarine volume/river discharge) is ~9 days (Engqvist, 1993). The deep waters of the estuary are confined by a small elevation (~30 m water depth) at its southern border (Brydsten, 1992, Fig. 1). The sediments consist of non-permeable silts and fine sands (Hellemann et al., 2017).

The Vistula estuary is part of the Polish Bay of Gdansk, southern Baltic Sea (Fig. 1), and covers an area of ~825 km², with a

volume of ~20 km³. It receives inputs from the Vistula River, whose mean discharge is 1080 m³ s⁻¹, resulting in a river plume of 0.5–12 m vertical and 4–30 km horizontal extent (Cyberska and Krzyminski, 1988). The mean water turnover time is ~6 days (calculated from data in Witek et al. 2003). Due to the absence of topographical restrictions, the Vistula estuary merges directly with adjacent coastal waters and the offshore waters of the Bay of Gdansk (Fig. 1). The border between the estuary and the latter occurs at ~50 m water depth, where the sediment changes from sand to silt, and can be recognized based on the

isotopic N-signature in the sediment that reflects the anthropogenic impact of the Vistula River (Thoms et al., 2018; Fig. 1). Water and sediment samples were taken from the Öre estuary (ÖE) and the Vistula estuary (VE) in spring and summer during four campaigns between 2014 – 2016 with the RV *Lotty* (ÖE I, ÖE II) and RV *Elisabeth Mann-Borgese* (VE I, VE II; Table 1). Water samples were obtained at three to six depths, from surface to bottom, using a rosette water sampler (5 L) connected to a conductivity-temperature-depth probe (CTD; VE I, II) or with Niskin bottles (5 L or 10 L; ÖE I, II) after the CTD cast.

Water samples from immediately above the sediment (20 – 40 cm) were taken from the overlying water of intact sediment cores. Sediment samples were collected using a Gemini twin corer (core iØ 8 cm, length 80 cm; silt, ÖE I, II), a multi-corer (core iØ 10 cm, length 60 cm; silt and fine sand, VE I, II), and a HAPS bottom corer (core iØ 14 cm, length 30 cm; sand, all campaigns) with a vibration unit (KC Denmark; vibration time 10–15 sec). Surface sediment slices (0 – 2 cm) were taken to determine basic sediment characteristics. Subsamples for denitrification rate measurements (n = 12 per site, except VE I: n =

20) and pore-water oxygen profiles (n = 3 per station) were collected in acrylic cores (iØ 2.3 cm, length 20 cm, except VE I: 15 cm), which were pushed gently into the sediment to fill 30% (silt) to 50% (sand) of each one and then closed under water without headspace. All samples were stored in the dark at in situ temperature until further processing within minutes (VE I, II) to maximum of 2 h (ÖE I, II) after sampling.



### 2.2 Environmental data

#### 2.2.1 Water column

Water column characteristics were measured with CTD-probes (Seabird-Scientific) as described by Hellemann et al. (2017; ÖE I, II) and Bartl et al. (2018; VE I, II). The extent of the BBL was determined based on density stratification (Bartl et al., 2018; Turnewitsch and Graf, 2003; Table S1). Water samples were considered to represent the BBL when the water sampler (0.5 – 1 m length) was completely inside the BBL during sampling. Concentrations of dissolved inorganic N nutrients ($NO_2^-$, $NO_3^-$, $NH_4^+$, defined as DIN) were measured colorimetrically following Grasshoff et al. (1999) and HELCOM guidelines (2014) using a continuous segmented flow analyser (QuAAtro, Seal Analytical; ÖE I, II) or as described in Bartl et al. (2018; VE I, II). Background subtraction of the colorimetric signals from Öre estuary samples was used to account for the water colour, which was altered due to the high dissolved organic matter content. Concentrations of chlorophyll $a$ (Chl.$a$) were measured using an optical sensor (Cyclops 7, Turner Designs) attached to a Seaguard CTD-probe (Aandeera; ÖE I, II) or with the fluorometric method (Edler, 1979; Wasmund et al., 2006, VE I, II). Particulate organic nitrogen and carbon (PON, POC) concentrations and the natural isotopic composition of POC (given as $\delta^{13}$C-POC) were measured using a continuous-flow isotope ratio mass spectrometer (Delta V Advantage) connected via an open split interface (ConFlo IV) to an elemental analyser (Flash 2000, all from Thermo Fisher Scientific) as described in Hellemann et al. (2017; ÖE I, II) and Bartl et al. (2018; VE I, II). The contribution of different POM sources to the total estuarine POM pool was estimated using a two-component mixing model (Goñi et al., 2003; Jilbert et al., 2018), described in detail in Hellemann et al. (2017), with terrestrial POM (C:N of 20) and phytoplankton-derived POM (C:N of 8) as end-members. Because high C:N ratios can also indicate degraded POM due to the preferential utilization of PON over POC (Savoye et al., 2003), a contribution of terrestrial POM can also represent degraded POM. The $\delta^{13}$C-POC value was used to distinguish between terrestrial and degraded POM; for the terrestrial POM of northern Baltic rivers $\delta^{13}$C-POC is below −28 ‰ (Rolff and Elmgren, 2000). The degradation state of POM was evaluated by determining the POC:Chl.$a$ and C:N ratios, which increase simultaneously during degradation (Savoye et al., 2003). POC:Chl.$a$ ratios < 200 indicate newly produced phytoplankton POM, and > 200 degraded POM (Cifuentes et al., 1988).

#### 2.2.2 Sediment characteristics

Sediment type classification using grain size distribution, as well as porosity and loss on ignition (LOI) using weight loss in drying are described in detail in Hellemann et al. (2017; ÖE I, II) and Thoms et al. (2018; VE I, II). The permeability ($K_m$) of the sand sediment in both estuaries was analysed using a constant head permeameter as described in Hellemann et al. (2017). Sediments with a $K_m \geq 2.5 \times 10^{-12}$ $m^2$ were considered permeable enough to enable advective pore-water flow with significant effects on sediment biogeochemistry in the Baltic Sea (Forster et al., 2003). Sediments with a $K_m < 2.5 \times 10^{-12}$ $m^2$ were defined as non-permeable, because here such effects are negligible.





### 2.2.3 Oxygen and ammonium in sediment pore waters

Oxygen pore-water concentration profiles were obtained at in situ temperature using Clark-type microelectrodes (ÖE I, II, VE I: 200 to 250μm vertical resolution, OX-100, Unisense; VE II coarse–fine sands: 500μm vertical resolution, OX-250, Unisense) as described in Hellemann et al. (2017). The oxygen penetration depth (OPD) in the sediment was determined from each profile, with the sediment surface identified by a characteristic break in the profile curve and additional visual estimates during profiling of the permeable sands. Profiles affected by fauna were discarded (max. 12–16 %).

Samples for the determination of pore-water $NH_4^+$ concentrations were taken from intact sediment cores, either by core slicing (resolution: 1 cm) under $N_2$ atmosphere followed by centrifugation and filtration (silts ÖE) or by using Rhizons$^{TM}$ (Rhizosphere Research Products; resolution: 1 cm for 1 to 5 cm depth, 2 cm for 5 to 11 cm depth; coarse silt and fine sand ÖE, VE) according to Thoms et al. (2018). The pore-water samples were immediately frozen and kept at −20°C until the analysis. Pore-water $NH_4^+$ concentrations were measured colorimetrically (Grasshoff et al., 1999), either manually using a spectrophotometer (UV-Vis 1201 LAMBDA2, Shimadzu, accuracy 5%; silt ÖE I, II) or automated using a continuous segmented flow analyser (QuAAtro, Seal Analytical, accuracy 5 – 10%; coarse silt and fine sand ÖE I, II, VE I, II). $NH_4^+$ concentrations were vertically integrated for the surface (0 – 2 cm) and the deeper (2 – 10 cm) sediment layer to yield total pools of pore-water $NH_4^+$ (μmol m$^{-2}$; Table 3).

## 2.3 Quantification of N-transformation processes

### 2.3.1 Nitrification and ammonium assimilation rates in the BBL

Water samples for $^{15}N$-$NH_4^+$ tracer incubations (Damashek et al., 2016; Ward, 2005) were collected from the BBLs of the two estuaries and processed as described in detail by Bartl et al. (2018). Briefly, six polycarbonate bottles were filled with water (100–170 mL from water overlying the sediment, 625 mL from the water column) and sealed gas-tight. The water samples were amended with $^{15}N$-$NH_4Cl$ (98 atom% $^{15}N$, Sigma Aldrich) to yield a sample enrichment of 0.05 μmol L$^{-1}$ (ÖE I, II, VE I) or 0.20 μmol L$^{-1}$ (VE II). One set of triplicates was filtered immediately through pre-combusted glass-fiber filters (GF/F Whatman, 3 h at 450°C), while the remaining triplicates were incubated for 5–7 h (ÖE I, VE I) or 3 h (VE II) in the dark at in situ temperature. Isotope dilution via ammonification during the incubation was minimized by keeping the incubation time short (Ward, 2011). The incubation was terminated by filtration; the filtrates and filters (rinsed with filtered seawater) were stored at −20°C until the analysis.

The $^{15}N$ content of $NO_3^-$+$NO_2^-$ in the filtrate was measured according to the denitrifier method (Casciotti et al., 2002; Sigman et al., 2001) using a continuous-flow isotope ratio mass spectrometer (IRMS, Delta V Advantage, Thermo Fisher Scientific) connected to a Finnigan GasBench II (calibration against the standards IAEA-N3 and USGS-34, accuracy: ± 0.14 ‰). Nitrification rates were calculated according to Veuger et al. (2013). Since the $^{15}N$ content of both, $NO_2^-$ and $NO_3^-$, is measured



simultaneously, the calculated nitrification rate is a bulk rate that includes $NH_4^+$ and $NO_2^-$ oxidation. Ammonium assimilation rates were calculated according to Dugdale and Wilkerson (1986) using the PON concentration and its $^{15}N$ content, measured from the filters as described above.

### 2.3.2 Gaseous N production in the sediment:

Benthic $N_2$ and $N_2O$ production in both estuaries was measured using the revised isotope pairing technique (r-IPT; Risgaard-Petersen et al., 2003), which accounts for the contributions of denitrification and anammox to total $N_2$ production. All non-permeable sediment samples from both estuaries were incubated using a diffusive set-up, in which the overlying water in the acrylic cores was enriched with $K^{15}NO_3$ (98 % $^{15}N$, Cambridge Isotope Laboratories) to final concentrations of 40, 80 and 120 µmol $L^{-1}$ (ÖE I, II, VE I; n = 4 per concentration, except VE I 120 µmol $L^{-1}$: n = 12; isotope enrichment in the water [Fn]: 84

– 100 %) or 30, 60, 90 and 120 µmol $L^{-1}$ (VE II; n = 3 per concentration; Fn: 86 – 100 %). The samples were subsequently incubated in the dark under gentle water mixing by magnetic stirrers at in situ temperatures for 3–5 h. The permeable sediment samples of VE II were also incubated using the above described diffusive set-up, since advective pore-water flow was most likely negligible during sampling (see sect. 4.1.3). The permeable sediment samples of VE I were incubated with an advective set-up, in which bottom water, enriched with $K^{15}NO_3$ (98 % $^{15}N$, Cambridge Isotope Laboratories) to final concentrations of

40, 80, and 120 µmol $L^{-1}$ (n = 5 – 7 per concentration; Fn range: 98 – 100%), was pumped through the oxic sediment layer. This layer was determined from oxygen profiles and used as an approximation of the sediment depth affected by advective pore-water flow (Supplement Fig. S1). The pumping rate (0.25 mL $min^{-1}$; IPC high-precision tubing pump, ISMATEC) at site-specific porosities led to pore-water velocities of ~7.6 cm $h^{-1}$. The tracer-enriched water was pumped from the top into the overlying water of the acrylic cores and drawn from the sediment at two opposing sides through pre-drilled holes (vertical

resolution 5 mm). This outflow was ~5 mm above the approximated oxic-anoxic interface, where denitrification takes place, to ensure that the flow reached this interface but did not affect deeper layers. In- and outflow ports were sealed with rubber plugs through which Tygon® tubing (ST R-3603/R-3607, iØ 2.3 mm) was inserted; all connecting interfaces were tightened with Teflon® tape. While the tracer-enriched water was pumped into the cores, the resident pore water within the advection-affected sediment layer could flow out; thereby, all water with contact to the estimated advective layer was exchanged with

tracer-enriched water within 2.5–3 h. Subsequently, one core per concentration was sampled, with the tubing of the remaining cores connected to a closed circulation set up for each core (Supplement Fig. S1) and incubated for ~5 h.

All incubations were stopped by mixing the sediment with the overlying water. After brief sediment settling, 12-mL subsamples were placed into gastight glass vials (Exetainer, Labco Scientific) with 0.5 mL of $ZnCl_2$ (100 % w/v, Merck). After a 5-mL Helium headspace had been created, the isotopic compositions of $N_2$ and $N_2O$ were analysed using a continuous-flow

IRMS (IsoPrime 100, Isoprime; standard gas: $N_2$, > 99.999 % purity, AGA) interfaced with a gas pre-concentrator system (TraceGas, Isoprime) and an automated liquid handler (GX-271, Gilson) at the Department of Environmental Sciences, University of Jyväskylä, Finland (VE I) or with a continuous-flow IRMS (Delta V Plus, Thermo Scientific, standard gas:



Oztech $N_2$, i.e. $\delta^{15}N$ vs air = -0.61, Oztech Trading Co.) interfaced with a gas bench and a pre-concentrator system (Precon, Thermo Scientific) at the Stable Isotope Facility, University of California, Davies, USA (VE II).

According to the r-IPT, a contribution of anammox to the measured $N_2$ production is indicated by a positive correlation between the production rate of $^{14}N$-$N_2$ (D14, calculated with the IPT, Nielsen, 1992) and an increasing $^{15}N$-$NO_3^-$ concentration in the incubations. In this case, calculation of denitrification rates follow Risgaard-Petersen et al. (2003). If D14 does not correlate positively with the $^{15}N$-$NO_3^-$ concentrations, denitrification is assumed to be the only process producing $N_2$ (Risgaard-Petersen et al. 2003) and calculations follow Nielsen (1992). Moreover, the valid application of both IPT and r-IPT requires a linear

dependency of the production rate of $^{15}N$-$N_2$ (D15) on increasing $^{15}NO_3^-$ concentrations. All dependencies were tested with a regression analysis (significance level: $p < 0.05$). Denitrification of $NO_3^-$ from the bottom water (Dw) and from nitrification within the sediment (Dn, coupled nitrification−denitrification) was calculated from D14 and the ratio of $^{15}N$-$NO_3^-$ to $^{14}N$-$NO_3^-$ in the water phase.

## 2.4 Statistical analyses

Significant differences between the factors 'site' (Öre estuary, Vistula estuary), 'season' (spring, summer) and 'sediment type' (permeable, non-permeable) were tested using the non-parametric Mann-Whitney U-test (2 factors, $n \geq 3$) or the non-parametric Kruskal-Wallis test (>2 factors, $n \geq 3$) combined with Dunn's post-hoc test (all SigmaPlot, version 13.0). Multivariate correlation analyses (Kendall's $\tau$, $n \geq 5$) were done between environmental variables and nitrification (ÖE II), ammonium assimilation (ÖE II, VE I, II), and denitrification (VE I, II) rates using SAS (version 9.4). No correlation analyses

were done for data of ÖE I because the sample size was too small ($n \leq 4$). In all analyses, the significance level was $p < 0.05$.

## 3 Results

### 3.1 Environmental variables

#### 3.1.1 Water column

In spring and summer, plumes of the Öre River and Vistula River were identified by their low potential density in the surface

water; vertical extent of $\leq 5$ m (Fig. 2, 3). The water column below the river plume was well mixed in spring and exhibited a thermohaline stratification in summer (Fig. 2, 3). In both estuaries, oxygen conditions differed seasonally but all water layers were oxic (> 230 µmol $L^{-1}$; Supplement Table S1). In spring, DIN concentrations were ~30 times higher in the Vistula River plume than in the Öre River plume, while concentrations in the BBL differed only by a factor of 2 (VE > ÖE; Fig. 2; Supplement Table S1). In summer, $NH_4^+$ concentrations in the BBL were higher in the Vistula estuary than in the Öre estuary

(Fig. 3), while the opposite was the case for $NO_3^-$+$NO_2^-$ (Supplement Table S1). In the Vistula estuary, PON and POC concentrations were highest in the surface waters in both seasons, while concentrations in the BBL were higher in summer than in spring (Fig. 2, 3; Table 2). In the Öre estuary, PON and POC concentrations were constant throughout the water column





in spring but were highest in the BBL in summer (Fig. 2, 3; Table 2). POC and PON concentrations were significantly higher in the BBL of the Öre estuary than of the Vistula estuary (Fig. 2, 3; Table 2).

In both seasons, POM in the Öre River and river plume contained a large share of terrestrial POM, while the Vistula River and river plume were dominated by phytoplankton-derived POM (Table 2). The terrestrial origin of POM from the Öre River and river plume was reflected by the high C:N ratios and low $\delta^{13}$C-POC values, neither of which occurred in the BBL of the Öre estuary nor in the Vistula River and its estuary (Table 2). In both estuaries and in both seasons, the estuarine POM contained a large share of phytoplankton-derived POM (Table 2), which was also reflected in the high Chl.$a$ concentrations measured

throughout the water column in spring and in the surface water in summer (Fig. 2, 3). POC:Chl.$a$ ratios in the two estuaries were < 200 throughout the water column in spring and > 200 in the BBL in summer, ranging from 250 to 1,200 in the Vistula estuary and from 980 to 12,000 in the Öre estuary (Fig. 4). The particulate C:N ratio of the surface water covered similar ranges in both estuaries and during both seasons (Fig. 4). In the BBL, particulate C:N ratios were significantly higher in the Öre (10.2 ± 0.9, n=9) than in the Vistula (8.6 ± 0.6, n=12) estuary in summer, but covered similar ranges in spring (ÖE: 7.1–

10.8, n=7 and VE: 6.6–13.5, n=18).

### 3.1.2 Sediment

Permeable sediments cover ~56% of the area of the Vistula estuary (Supplement Fig. S2), while the sediments in the Öre estuary are non-permeable. LOI differed significantly between permeable and non-permeable sediments but not between estuaries or seasons (Table 3). The oxygen profiles in the permeable sediments of the Vistula estuary were sigmoidal with

nearly constant oxygen concentrations in the top millimetres of sediment in spring, and nearly parabolic in summer, similar to the profiles of the non-permeable sediments in both seasons (Fig. 5). The mean OPD in the permeable sediments was 60% lower in summer than in spring and did not differ significantly from the OPD in the non-permeable sediments of the same season (Table 3). Pore-water $NH_4^+$ pools differed seasonally only in the permeable sediments of the Vistula estuary, with ~73% (surface sediment layer) and ~37% (deep sediment layer) more $NH_4^+$ in summer than in spring. The deep $NH_4^+$ pool of the

non-permeable sediments was significantly higher in the Vistula than in the Öre estuary in both seasons (Table 3).

### 3.2 Nitrogen transformation processes

### 3.2.1 Nitrification and ammonium assimilation in the BBL

Nitrification rates in the BBL did not significantly differ either between seasons or between estuaries (Table 4). In both estuaries, nitrification rates correlated positively with PON and POC concentrations in summer (VE: Kendall's τ=0.81, p=0.01,

n=7 [Bartl et al., 2018]; ÖE: Kendall's τ=0.71, p=0.02, n=7; Fig. 6A). In contrast to the Vistula estuary, nitrification rates in the BBL of the Öre estuary showed a negative trend with the particulate C:N ratio (Fig. 6B). Ammonium assimilation rates in the BBL differed seasonally (spring < summer) and were higher in the Öre than in the Vistula estuary in spring but not in summer (Table 4). In both estuaries, the rates correlated positively with PON and POC concentrations (VE: Kendall's τ=0.61,




p=0.02, n=9; ÖE: Kendall's $\tau$=0.71, p=0.02, n=7; Fig. 6C), while in the Öre estuary the rates correlated negatively with the C:N ratio (Kendall's $\tau$=-0.71, p=0.02, n=7; Fig. 6D).

### 3.2.2 Denitrification in the sediment

No anammox was found at any of the sites, indicating that $N_2$ production in the studied estuaries originated entirely from denitrification. $N_2O$ production during denitrification was $\leq 1.8\%$ of total $N_2$ production in all samples, and denitrification rates were expressed as the sum of $N_2 + N_2O$. In the Öre estuary, denitrification was not detectable in spring. In the Vistula estuary, rates in spring were $\geq 60\%$ lower than in summer, with the rates in the permeable sediment being half of the rates in the non-permeable sediment (Table 4). In summer, denitrification rates in the Vistula estuary did not differ between permeable and non-permeable sediments and were more than twice as high as in the Öre estuary (Table 4). Denitrification in both estuaries was primarily coupled to nitrification in the sediment (Dn $\geq$ 93 %). In summer, coupled nitrification-denitrification correlated positively with LOI in the surface sediments of the Vistula estuary (Kendall's $\tau$=0.73, p=0.04, n=6; Fig. 6E; one non-permeable site excluded), but not in the Öre estuary. Coupled nitrification-denitrification correlated negatively with the particulate C:N ratio in the Öre (Kendall's $\tau$=0.80, p=0.05, n=5; Fig. 6F) but not the Vistula estuary. The particulate C:N ratio in the BBL was used in the correlation analysis because it is similar to the C:N ratio of the surface sediment, of which the sample size did not suffice for statistical analyses (data not shown).

## 4 Discussion

### 4.1 Environmental settings of the Vistula and Öre estuaries

#### 4.1.1 Site-specific and seasonal environmental settings

Compared to the oligotrophic Öre estuary, the significantly higher concentrations of DIN in the spring river plume of the eutrophied Vistula estuary reflected the 30 times higher discharge and 3 orders of magnitude higher DIN load of the Vistula river (Table 1). The high nutrient availability in the surface water of the Vistula estuary can support primary production rates ~6 times higher than those in the nutrient-limited Öre estuary (Ask et al., 2016; Witek et al., 1999), as evidenced by the higher concentrations of PON, POC and Chl.$a$ in the surface water (Table 2; Fig. 2, 3).

Interestingly, summertime $NO_3^-+NO_2^-$, PON and POC concentrations in the BBL were significantly higher in the Öre than in the Vistula estuary (Supplement Table S1, Table 2) which may have resulted from accumulation of mineralization educts (PON, POC) and products ($NO_3^-+NO_2^-$). Indeed, using the average BBL nitrification rate from April 2015 (21 nmol $L^{-1}$ $d^{-1}$) and assuming, that it was constant for the three following months, nitrification in the BBL produced as much as ~2.0 µmol $NO_3^-+NO_2^-$ $L^{-1}$ which is close to the average concentration measured in August 2015 (Supplement Table S1). The accumulation of elements in the benthic system is favoured by the basin-like bottom topography of the Öre estuary and restricted bottom-water exchange, which allows a long particle residence time of more than one year (Brydsten and Jansson,




1989). By contrast, the open shape and unrestricted bottom topography of the Vistula estuary allows the free exchange of
bottom-water with adjacent coastal and offshore waters. Consequently, there were no signs of accumulation in this estuary.

Among the factors that define the trophic state of an ecosystem is the long-term input of organic matter (Nixon, 1995), visible
in the elemental pools of sediments. This was particularly pronounced in the non-permeable sediments of the Vistula estuary,
where deep $NH_4^+$ pools were twice as large as in the oligotrophic Öre estuary. The small $NH_4^+$ pools of the permeable
sediments, found only in the Vistula estuary, were an exception to this pattern, since high elemental turnover based on
advective pore-water flow results in the limited accumulation of organic matter (Boudreau et al., 2001; see sect. 4.1.3).

While the two studied estuaries differ in size, bottom topography, sediment type and trophic state, they were similar with
respect to the seasonal density stratification and the distribution pattern of DIN and POM. In spring, riverine DIN loads in the
estuaries had no direct contact with the respective benthic system but were likely taken up during primary production by the
developing spring blooms (Fig. 2). The POM generated in the surface water could have been exported from the estuaries or
have reached the benthic system via close benthic-pelagic coupling. The elevated Chl.*a* concentrations throughout the water
columns of the Vistula and Öre estuaries (Fig. 2) suggested that a part of the estuarine phytoplankton was dispersed down to
the aphotic benthic system. Hence, riverine DIN supply to the benthic system is indirect via POM build-up and sedimentation.
Thereby, it is uncoupled from the timing of peak river discharge while it may impact the benthic system later in the year
(Hellemann et al., 2017).

**4.1.2 Characterization of POM in the Öre and Vistula estuaries**

In spring, the high variability in the C:N and POC:Chl.*a* ratios in the BBL of the Vistula estuary suggested that its POM was
a mixture of newly produced phytoplankton POM (POC:Chl.a < 200), degraded phytoplankton POM from the previous year
(POC:Chl.*a* > 200 at C:N < 12), and terrestrial POM (C:N > 12; Fig. 4, Table 2). While terrestrial POM accounted for a large
share of the POM in the Öre River, its contribution to the POM in the BBL of the Öre estuary in spring was small (Fig. 4,
Table 2). This was likely due to the abundant, widely dispersed estuarine phytoplankton (Fig. 2) and the immediate
sedimentation of riverine particles close to the river mouth (Forsgren and Jansson, 1992), such that the terrestrial POM of the
Öre River might have not reached the benthic system of the sampled stations during the field campaign.

While the C:N and POC:Chl.*a* ratios in the BBL of the Vistula estuary in summer indicated POM degradation, the ratios were
significantly lower than in the Öre estuary (Fig. 4). High riverine POM loads (Table 1) and high estuarine primary production
rates in the Vistula estuary (Witek et al., 1999) may have resulted in the continuous input of phytoplankton-derived POM into
the benthic system. In the Öre estuary, by contrast, inputs of phytoplankton-derived POM are lower and summertime
POC:Chl.*a* and C:N ratios in the BBL (Fig. 4) suggested that this POM was sequestered during the spring season, leaving more
degraded and N-depleted POM in the benthic system in summer (Hellemann et al., 2017).




### 4.1.3 Permeable sediments of the Vistula estuary

Permeable sediments can have significant advective pore-water flow, preventing accumulation of organic matter in the sediment (Huettel et al., 2014). In the permeable sediments of the Vistula estuary in spring, advective pore-water flow was indicated by the sigmoidal shape of the oxygen profiles (Revsbech et al., 1980; Fig. 4) and the low pore-water $NH_4^+$ pools, similar to the results from subtidal permeable sediments of the North Sea (Ehrenhauss et al., 2004; Lohse et al., 1996). Low $NH_4^+$ pore-water pools can result from the high consumption of $NH_4^+$ via nitrification in the large oxic sediment volume and/or

enhanced $NH_4^+$ release from the sediment through advection (Huettel et al., 1998). However, the significantly higher $NH_4^+$ pools in summer than in spring suggested $NH_4^+$ accumulation in the permeable sediments. Together with the nearly parabolic shape of the oxygen profiles and shallow OPDs this indicated diffusive mass transport in summer (Fig. 5, Revsbech et al., 1980) rather than the generally anticipated advective transport in those sediments. Seasonally differing oxygen profiles also characterized permeable sediments of the German Bight, North Sea (Lohse et al., 1996), where the parabolic, shallow profiles

in summer have been attributed to the "absence of a turbulent water column". The authors observed that oxygen consumption in the sediment can distort the shape of originally advective (sigmoidal) oxygen profiles within 30 – 60 min at a diffusive oxygen uptake (DOU) rate of 6.7 mmol $m^{-2}$ $d^{-1}$. In the permeable sediments of the Vistula estuary, oxygen profiles were measured within ~30 min of the first sampling and showed a much lower summertime DOU ($0.6 \pm 0.3$ mmol $m^{-2}$ $d^{-1}$, n = 21) than in the German Bight (Lohse et al., 1996). Hence, it is unlikely that the very low DOU in this study created artificial

diffusive profiles over this time span. Instead, pressure gradients at the surface of the permeable sediments in the Vistula estuary in summer may not have sufficed to facilitate significant advective pore-water flow. Such pressure gradients mainly form via the interaction of near-bottom flow and bottom topography or by waves (Santos et al., 2012). We used modelled near-bottom flow velocity data for our sampling period to examine whether the theoretical interaction of that flow with a topographic object could create pressure gradients sufficient to drive advection (see Supplements). The modelled near-bottom flow velocity

was very low ($< 2.5$ cm $s^{-1}$), probably due to the pronounced thermohaline stratification, and resulted only in minor pressure gradients ($< 0.15$ Pa) on a 3 cm-high mound (Table S2). The resulting Peclet number was below the threshold for pore-water advection within the sediment ($\geq 5$, Bear, 1972; Table S2). We therefore suggest that the pressure gradients at the permeable sediment surface of the Vistula estuary in summer 2014 were too low to create significant advective pore-water flow. This leaves diffusion and fauna-induced fluxes as the main transport processes during that time, leading to a temporary accumulative

character of the permeable sediments. Presumably, this temporary switch between transport regimes is more likely to occur in low-energy environments, such as the nearly non-tidal Baltic Sea. Further research is needed to evaluate the frequency of such changes and their impact on biogeochemical processes.



### 4.2 The effect of contrasting environmental settings on benthic microbial N-turnover

#### 4.2.1 Nitrification and ammonium assimilation in the BBL

Pelagic nitrification rates are often higher in eutrophied than in oligotrophic estuaries, due to the increased availability of $NH_4^+$ and POM (Bianchi et al., 1999; Dai et al., 2008; Damashek et al., 2016). However, this was not the case in the two studied Baltic estuaries, where nitrification rates were in the same range in the BBL of the eutrophied Vistula estuary and the oligotrophic Öre estuary (Table 4). Likewise, a previous study showed similar gene and transcript abundances as well as a similar composition of ammonium-oxidizing archaea and bacteria in the BBLs of these two estuaries (Happel et al., 2018). In

both estuaries, higher nitrification activities in summer were anticipated based on the higher concentrations of $NH_4^+$ and POM than in spring; however, the rates did not differ seasonally. For the Vistula estuary, this has been attributed to the intense competition for the substrate ammonium with heterotrophic ammonium assimilation (Bartl et al., 2018). The same can be assumed for the Öre estuary, where, furthermore, a nitrifier community with seasonally similar characteristics has been described (Happel et al., 2018). The nitrification rates from the Baltic coast measured in this study were similar to previously

reported pelagic rates from the Baltic Proper (0−84 nmol $L^{-1}$ $d^{-1}$ at 80–117 m water depth, Hietanen et al., 2012), suggesting a Baltic Sea-specific range of nitrification rates. Globally, these are in the lower range of the highly variable bottom-water rates from other coastal zones (e.g., 0–20 nmol $L^{-1}$ $d^{-1}$, Heiss and Fulweiler, 2016; 150–494 nmol $L^{-1}$ $d^{-1}$, Bristow et al., 2015; 10–310 nmol $L^{-1}$ $d^{-1}$, Damashek et al., 2016, 32–4600 nmol $L^{-1}$ $d^{-1}$, Hsiao et al., 2014).

Because ammonium assimilation is a temperature-dependent process (Baer et al., 2014; Hoch and Kirchman, 1995), higher

temperatures in summer enhanced ammonium assimilation rates in both estuaries. In spring, ammonium assimilation rates in the BBL were higher in the Öre than in the Vistula estuary, a finding attributable to the heterotrophic character of the Öre estuary (Sandberg et al., 2004; Wikner and Andersson, 2012). The ammonium assimilation rates measured in this study represent typical estuarine rates, as they are similar to heterotrophic ammonium assimilation rates determined in the surface waters of the Delaware estuary (13–930 nmol $L^{-1}$ $d^{-1}$; Hoch and Kirchman, 1995) and in the bottom-waters at the Washington

coast (500 nmol $L^{-1}$ $d^{-1}$; Ward et al., 1984).

In the BBL in summer, POM was the main factor controlling nitrification and ammonium assimilation in the Vistula and Öre estuaries. The positive correlations between the rates and concentrations of PON and POC likely resulted from the particle-attachment of nitrifiers (Karl et al., 1984; Phillips et al., 1999) and ammonium-assimilating bacteria, as both groups may benefit from the direct $NH_4^+$ supply released from PON degradation (Bartl et al., 2018; Dang and Chen, 2017; Hsiao et al.,

2014; Klawonn et al., 2015). Furthermore, recent studies found nitrifying species capable of degrading organic matter to obtain $NH_4^+$ (Alonso-Sáez et al., 2012; Kuypers et al., 2018), which in our study may have contributed to the positive correlation between nitrification rates and PON. Interestingly, at higher PON concentrations, the increase in nitrification and ammonium assimilation was stronger in the Vistula than in the Öre estuary (Fig. 6), because the less-degraded POM in the former (see sect. 4.1.2) contains more organic N as an indirect substrate source for nitrification. By contrast, the more degraded POM in





the Öre estuary leads to a limited availability of organic N as a potential $NH_4^+$ source, as further reflected in the negative correlation between nitrification and ammonium assimilation rates and the C:N ratio (Fig. 6). In summary, the magnitude of nitrification and ammonium assimilation in the BBL was not influenced by the different trophic state or by seasonal differences. However, the regulation of those two processes differed depending on the trophic state, i.e., the availability of organic N from POM.

**4.2.2 Denitrification in the sediment**

Heterotrophic denitrification uses dissolved organic C and $NO_3^-$ in a ratio of 1:1 (Taylor and Townsend, 2010), which shows the potential of organic C to limit denitrification activity. Differences in denitrification rates between the estuaries and between seasons were related to the availability of organic C in the respective benthic system, which in both estuaries originated mainly from phytoplankton-derived POM. The average summer denitrification rate was more than 2-fold larger in the Vistula estuary

(~360 µmol N m$^{-2}$ d$^{-1}$) than in the Öre estuary and similar to the summertime rates in other eutrophied estuaries of the Baltic Sea (320–360 µmol N m$^{-2}$ d$^{-1}$, Bonaglia et al., 2014; 90–910 µmol N m$^{-2}$ d$^{-1}$, Silvennoinen et al., 2007; 290–350 µmol N m$^{-2}$ d$^{-1}$, Nielsen and Glud, 1996).

The seasonal difference in denitrification rates (spring < summer) was related to the limited denitrification activity in spring, which most likely reflected the low availability of labile organic C (Bradley et al., 1992; Hellemann et al., 2017; Taylor and

Townsend, 2010), as also found in other aphotic coastal sediments (Hietanen and Kuparinen, 2008; Jäntti et al., 2011). While newly produced phytoplankton POM was present in both benthic systems during the spring samplings (Fig. 4), low bottom-water temperatures (Supplement Table S1) likely slowed its degradation to dissolved C components suitable for denitrification. Coupled nitrification-denitrification (Dn) in the Vistula estuary increased significantly with increasing amounts of organic matter, probably because of the large share of easily degradable POM containing sufficient organic C and N sources for

denitrification, as reported previously (e.g. Deutsch et al., 2010; Finlay et al., 2013; Jäntti et al., 2011; Seitzinger and Nixon, 1985). However, this was not the case in the Öre estuary, where the more degraded state of the POM (Fig. 4) reduced the availability of organic N and C as substrates as evidenced by the negative correlation between denitrification and the particulate C:N ratio (Fig. 6F). Hence, POM quality is a key factor controlling denitrification (Eyre et al., 2013; Hietanen and Kuparinen, 2008), especially in oligotrophic systems (Hellemann et al., 2017).

In both estuaries and during both seasons, denitrification mainly used $NO_3^-$ from nitrification in the sediment, not $NO_3^-$ from the BBL, which is common in coastal sediments with sufficiently deep OPD (> 1 mm) and low $NO_3^-$ concentrations in the BBL (Rysgaard et al., 1994). Hence, only a small fraction (< 10%) of the $NO_3^-$ from the BBL was removed by denitrification, indicating a weak coupling of BBL and sediment in terms of N removal. This was also true for the permeable sediments under advective pore-water flow in the Vistula estuary. In those types of sediments, the dominance of the $NO_3^-$ source is controversial



(Kessler et al., 2013; Gihring et al., 2010; Marchant et al., 2016; Rao et al., 2007) and depends on, for example, pore-water velocity and the sediment respiration rate (Kessler et al., 2012).

### 4.2.3 Evaluation of rate measurements in the permeable Vistula sediment

The permeable sediments along the southern coast of the Baltic Sea have been suggested to account for substantial N removal (Korth et al., 2013; Voss et al., 2005a), similar to the permeable sediments in the North Sea and Atlantic Bight (Gao et al.,
2012; Rao et al., 2007). In this study, the permeable sediments of the Vistula estuary were subjected to advective pore-water flow in spring; thus, denitrification rates were accordingly obtained by using an advective incubation design. These rates were lower than those of non-permeable sediments, but this may have been due to the limitations of our incubation design in representing advective flow. We used the OPD as an estimate of the sediment depth affected by the pore-water flow. This is an approximation, as oxygen profiles are the product of the balance between oxygen supply and consumption (Ziebis et al.,
1996). Pore-water flow in our incubations was similar to that of high-energy waters (Huettel et al., 1996; Precht et al., 2004) and was therefore probably too high to realistically represent Baltic Sea conditions. Over the course of the incubation, the flow increased the OPD in most of the investigated cores (data not shown) and the subsequent oxygenation of formerly anoxic sediment layers shifted the oxic-anoxic interface downwards. The delay until the microbial community adapted to the new conditions may have reduced denitrification activity, which would explain the measured low rates. Nonetheless, it is unlikely
that in situ denitrification rates in the permeable sediment would have been significantly higher than those measured in the non-permeable sediment, due to the low availability of usable dissolved organic C in spring. A realistic representation of advective pore-water flow in biogeochemical incubations remains challenging (Huettel et al., 2014) and needs further technical development.

Based on the summer samples, permeable sediments in the Vistula estuary were not subjected to significant advective pore-
water flow, thus allowing the use of a diffusive incubation design. The obtained denitrification rates were similar to those of the non-permeable sediment (Table 4), possibly due to the same diffusive substrate supply and the same depth of the denitrification layer. As our results are from a single summer season, further research is needed to evaluate the role of permeable sediments in denitrification in the Baltic coastal zone.

### 4.2.4 N-removal as an indicator of coastal filter efficiency

The efficiency of the coastal N filter is often evaluated by estimating the N removal efficiency (e.g. Asmala et al., 2017; Deek et al., 2013; Khalil et al., 2013). This is done by extrapolating denitrification rates to the entire estuarine sediment area and time (here: sampling month), divided by the riverine TN load. Despite their significantly different rates, the estuaries each removed ~5% of the riverine TN loads in the respective summer months. This amount is at the lower end of the estimated removal efficiencies of temperate estuaries (3–26%; Deek et al., 2013; Fear et al., 2005; Jäntti et al., 2011; Silvennoinen et al.,
2007; Seitzinger and Nixon, 1985). Asmala et al. (2017) estimated that ~16% of the riverine TN load entering the Baltic Sea



is removed by coastal denitrification, with most of the removal occurring in lagoons, and concluded that the Baltic Sea coastal zone is an inefficient N filter. However, based on isotopic data and long-term nutrient concentrations, Voss et al. (2005a, 2011) suggested that most of the riverine N is sequestered and removed within the Baltic coastal zones. The anticlockwise circulation pattern in the Baltic Sea, resulting in alongshore coastal jets and restricted cross-shore mixing (Radtke et al., 2012), may

support coastal N retention. In this case, the coastal N filter efficiency would depend on the transport and storage of riverine N within the Baltic coastal zone, providing time for N retention processes to recycle N until its eventual permanent removal. Accordingly, N removal alone, e.g., via denitrification rates, relative to riverine TN loads may not be a good indicator of the N filter efficiency in river dominated coastal zones.

### 4.3 Key drivers of the coastal N-filter

In stratified estuaries, phytoplankton-derived POM is the essential link between land-derived DIN inputs in surface waters and the spatially and temporally separated N filter of the benthic system. Through close benthic-pelagic coupling, phytoplankton-derived POM functions as a carrier and temporary reservoir of organic N and C that controls benthic nitrification, ammonium assimilation, and denitrification rates. This link was recently suggested for the Öre estuary with respect to denitrification (Hellemann et al., 2017) and may also apply for the Vistula estuary.

We estimated the amount of riverine DIN taken up by primary production generating phytoplankton-derived POM. In the Öre estuary in April 2015, N uptake for the entire estuarine area (71 km$^2$) was calculated using an estuarine primary production rate of 0.39 g C m$^{-2}$ d$^{-1}$ (DBotnia, 2016) and the Redfield C:N ratio of 6.6. The resulting areal N uptake rate of 4.2 t d$^{-1}$ was an order of magnitude higher than the riverine DIN load in April 2015 (0.53 t d$^{-1}$); thus, all riverine DIN may have been taken up by estuarine phytoplankton. The high share of phytoplankton-derived POM in the BBL in spring suggested that a

considerable amount of this easily degradable POM reached the benthic system (Fig. 7). Over a particle residence time of more than one year (Brydsten and Jansson, 1989), N may undergo cycles of retention via ammonification, nitrification, re-assimilation to PON, and DNRA before it is eventually removed via coupled nitrification-denitrification in the sediment (Hellemann et al., 2017; Fig. 7). This is likely enabled by the geomorphology of the Öre estuary, such that even at low rates of nitrification and denitrification the estuary is an effective coastal N filter.

In the Vistula estuary, primary production rates, estimated from the riverine DIN load for March 2016 (453 t d$^{-1}$), would need to be as high as 3.6 g C m$^{-2}$ d$^{-1}$ to consume all riverine DIN. However, primary production rates in spring range from 0.3 to 2.8 g C m$^{-2}$ d$^{-1}$ (March-May, Voss et al., 2005b; Witek et al., 1999). Accordingly, the effective uptake of riverine DIN depends on the co-occurrence of peaks in river discharge and in primary production during the spring season. Furthermore, the open shape of the estuary and its unrestricted bottom topography may well enable the transport of riverine DIN and suspended

estuarine POM out of the estuary and along the coastal zone throughout the year (Voss et al., 2005b). We thus hypothesize that the coast-parallel transport of nutrients and estuarine POM extends the estuarine filter of the Vistula estuary to the adjacent

coastal zones (Fig. 7), where microbial N retention and N removal could take place over a larger area and a longer time scale. However, this remains to be determined in future research that takes into account the transport and mixing processes in open river-dominated coastal systems, such as the Vistula estuary, along with microbial N retention and N removal quantifications.

## 5 Conclusion

Contrary to our expectations the different trophic states of the Vistula and Öre estuaries influenced only the denitrification rates in the sediment, not the rates of ammonium assimilation and nitrification in the BBL. However, all three processes were shown to depend on the availability of easily degradable phytoplankton-derived POM as a substrate source, due to the stratification-driven separation of riverine DIN loads and the benthic system. Phytoplankton-derived POM is not only a carrier of bioavailable N but also functions as a temporary N reservoir through long particle residence times or coast-parallel transport. Hence, the efficiency of a coastal filter may depend not only on the rates of microbial N retention and N removal, but also on the geomorphological and hydrological features affecting the residence time and availability of degradable POM in coastal benthic systems. A temporary storage of POM within the coastal zone, on the other hand, may foster coastal eutrophication, thereby potentially decreasing the coastal filter function by, e.g., reducing N removal under conditions of evolving hypoxia (Jäntti and Hietanen, 2012). Especially in southern Baltic coastal zones, where riverine TN loads are consistently high (HELCOM, 2018), the coastal filter capacity may have reached its limits. Hence, the reduction of anthropogenic N inputs in Baltic catchment areas remains one of the most important goals of the Baltic Sea community.

**Author contribution**

Ines Bartl and Dana Hellemann: lead in Conceptualisation, Investigation, Formal analysis, Visualisation, Writing – original draft, review and editing
Christophe Rabouille, Kirstin Schulz, Petra Tallberg: Supporting Investigation, Writing – review and editing
Susanna Hietanen and Maren Voss: Supporting Conceptualisation, Funding acquisition, Supporting Investigation, Resources, Writing – review and editing

**Competing interests**

The authors declare that they have no conflict of interest.

**Acknowledgements**

We thank the participants of the field campaigns EMB077, EMB123, Öre I and II, and especially the captain and crew of the R/V *Elizabeth-Mann-Borgese*, and Daniel Conley for facilitating the sampling campaigns in the Öre estuary. The Umeå Marine



Sciences Center provided valuable marine infrastructure, environmental monitoring data, and laboratory support. Thanks to
Iris Liskow, Christian Burmeister, Aisha Degen-Smyrek, Sanni Aalto, Samu Elovaara, Anni Jylhä-Vuorio, Natalia Kozak, Bruno Bombled, Laetitia Leroy, Niels van Helmond, and Wytze Lenstra for their dedicated support in the field and in the lab. This project was supported by the BONUS COCOA project, funded jointly by the European Union, the Academy of Finland (grant agreement 2112932-1) and the German BMBF (grant number 03F0683A), as well as the Chancellor's travel grant of the University of Helsinki and the Academy of Finland (projects 272964, 303774 and 267112). Funding for Kirstin Schulz
was provided by the Dutch STW project "Sediment for the salt marshes: physical and ecological aspects of a mud motor" (grant number 13888).

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





## Tables

**Table 1: Sampling details of the field campaigns, as well as river discharge and nitrogen (N) loads during the sampling months, and the annual average.**

| Site | Cruise | Date | Season | River discharge[a] | TN load | DIN load | PON load |
|------|--------|------|--------|--------------------|---------|----------|----------|
|      |        |      |        | ($m^3 s^{-1}$) | (t month$^{-1}$) | (% of TN) | (% of TN) |
| Öre estuary | ÖE I | 20–24 April 2015 | Spring | 66 | 98 | 17 | 29 |
|  | ÖE II | 03–07 August 2015 | Summer | 26 | 26 | 3 | 22 |
|  |  |  | Annual average | 36 | 36[b] | 16[b] | 26[c] |
| Vistula estuary | VE I | 28 February–10 March 2016 | Spring | 1500 | 16172 | 87 | 6 |
|  | VE II | 04–15 July 2014 | Summer | 932 | 2621 | 3 | 10[e] |
|  |  |  | Annual average | 1080 | 8100[d] | 63[d] | 8[f] |

TN = total N, DIN = dissolved inorganic N, PON = particulate organic N, n.a. = not analyzed.

[a] Öre River: www.vattenwebb.smhi.se (annual average: 2004–2014); Vistula River: annual average discharge (1951-1990; Pastuszak and Witek, 2012); discharge of VE I and VE II from Polish national monitoring by the Institute of Meteorology and the Water Management National Research Institute

[b] http://miljodata.slu.se/mvm/ (1967–2014, without 1975)

[c] Average of spring and summer

[d] Average loads from Pastuzak and Witek (2012; period 1988-2011) and from Polish national monitoring by the Institute of Meteorology and the Water Management National Research Institute (period 2014-2015)

[e] Stepanauskas et al. (2002)

[f] Average of spring and summer






**Table 2: Concentration of particulate organic carbon (POC) and nitrogen (PON); natural isotopic composition of POC (δ13C-POC); the contribution of terrestrial and phytoplankton-derived particulate organic matter (POM) to the total POM pool measured in the river and river plume water as well as at the surface and in the bottom boundary layer (BBL) of the Öre and Vistula estuaries in spring and summer. The contribution of POM sources was estimated based on a two-component mixing model following Jilbert et al. (2017), using end members from Goñi et al. (2003). Values are average and standard deviation of each water layer. The number of replicates is shown in parentheses.**

| Site | Season | Water source | POC (µmol L$^{-1}$) | PON (µmol L$^{-1}$) | δ$^{13}$C-POC (‰) | Contribution terrestrial POM (%) | Contribution phytoplankton POM (%) |
|---|---|---|---|---|---|---|---|
| Öre estuary[a] | Spring | River | 153.6 | 11.2 | -29.1 | 71 | 29 (1) |
| | | River plume | 53.7 | 5.1 | -29.5 | 44 | 55 (1) |
| | | Surface | 40.2 ± 13.5 (8) | 4.3 ± 1.4 (8) | -25.7 ± 1.0 (8) | 19 ± 16 | 83 ± 16 (8) |
| | | BBL | 38.3 ± 8.7 (9) | 4.4 ± 0.9 (9) | -25.0 ± 1.1 (10) | 19 ± 16 | 81 ± 16 (10) |
| | Summer | River | 67.2 | 5.7 | -30.2 | 56 | 44 (1) |
| | | River plume | 46.9 ± 0.7 (3) | 4.1 ± 0.7 (3) | -28.7 ± 0.2 (3) | 55 ± 16 | 45 ± 16 (3) |
| | | Surface | 34.1 ± 7.9 (13) | 4.0 ± 0.8 (13) | -26.5 ± 0.6 (13) | 15 ± 11 | 85 ± 11 (13) |
| | | BBL | 28.0 ± 7.0 (17) | 3.3 ± 0.7 (17) | -26.1 ± 0.4 (9) | 38 ± 11 | 62 ± 11 (9) |
| Vistula estuary[b] | Spring | River | 164.2 | 16.5 | -25.7 | 37 | 63 (1) |
| | | River plume | 61.1 ± 25.9 (8) | 6.9 ± 2.5 (8) | -26.5 ± 1.4 (8) | 25 ± 14 | 75 ± 14 (8) |
| | | Surface | 45.6 ± 15.8 (6) | 5.8 ± 2.4 (6) | -24.8 ± 0.7 (6) | 10 ± 16 | 90 ± 16 (6) |
| | | BBL | 18.7 ± 7.6 (18) | 2.7 ± 1.2 (18) | -25.6 ± 0.8 (17) | 31 ± 24 | 69 ± 24 (18) |
| | Summer | River | - | - | - | - | - |
| | | River plume | 103 | 10.2 | -25.8 | 33 | 67 (1) |
| | | Surface | 73.6 ± 34.6 (7) | 8.3 ± 3.7 (7) | -25.7 ± 0.6 (7) | 20 ± 10 | 80 ± 10 (7) |
| | | BBL | 33.5 ± 11.9 (15) | 4.4 ± 1.3 (15) | -25.5 ± 0.8 (9) | 15 ± 10 | 85 ± 10 (9) |

[a] Including data from Hellemann et al. (2017)
[b] Including POC and PON concentrations from Bartl et al. (2018)



**Table 3: Sediment characteristics in the Öre and Vistula estuaries in spring and summer. Porosity and LOI are from a sediment layer depth of 0–2 cm. Depth, porosity and LOI are reported as ranges per season; all other values are reported as the average and standard deviation. The number of replicates is shown in parentheses.**

| Site | Season | Sediment | Depth (m) | $K_m$ ($10^{-12}$ m$^2$) | Sediment type | $\phi$ | LOI (dw %) | OPD (mm) | NH$_4^+$ surface pool (μmol m$^{-2}$) | NH$_4^+$ deep pool (μmol m$^{-2}$) |
|---|---|---|---|---|---|---|---|---|---|---|
| Öre estuary[a] | Spring | Non-permeable | 18–37 | 0.1 ± 0.1 (2) | Silt (Sandy) very coarse silt (Silty) very fine sand | 0.6–0.9 (7) | 1.9–12.8 (7) | 7.2 ± 0.9 (13) | 360 ± 232 (3) | 4743 ± 1845 (6) |
| | Summer | Non-permeable | 18–34 | 0.2 ± 0.1 (2) | Silt (Silty) very fine sand (Silty) fine sand | 0.6–0.9 (7) | 1.5–8.3 (7) | 3.5 ± 0.9 (38) | 473 ± 309 (7) | 4079 ± 2331 (7) |
| Vistula estuary[b] | Spring | Permeable | 22–36 | 6.9 ± 3.6 (7) | Fine sand Medium sand | 0.4–0.5 (8) | 0.3–1.3 (8) | 10.1 ± 4.5 (40) | 92 ± 48 (4) | 2899 ± 1103 (4) |
| | | Non-permeable | 16–59 | - | (Silty) very fine sand Fine sand | 0.4–0.8 (3) | 1.2–4.9 (3) | 3.2 ± 0.9 (21) | 428 ± 173 (2) | 15 362 ± 5996 (2) |
| | Summer | Permeable | 25–49 | 9.0 ± 8.1 (5) | Fine sand Medium sand Coarse sand | 0.3–0.4 (5) | 0.6–2.3 (5) | 4.1 ± 1.3 (20) | 336 ± 183 (5) | 4596 ± 1432 (5) |
| | | Non-permeable | 17–50 | 0.7 ± 0.2 (2) | Very fine sand Fine sand | 0.5–0.6 (3) | 2.4–11.6 (3) | 3.2 ± 1.2 (13) | 574 ± 284 (3) | 11 422 ± 7108 (3) |

Km = sediment permeability, $\phi$ = porosity, LOI = loss on ignition, OPD = oxygen penetration depth, NH$_4^+$ surface pool = ammonium concentration integrated over a sediment depth of 0–2 cm, NH$_4^+$ deep pool = ammonium concentration integrated over a sediment depth of 2–10 cm.
[a] Including data from Hellemann et al. (2017)
[b] Sediment type, porosity, LOI, and pore water NH$_4^+$ concentrations from Thoms et al. (2018)




**Table 4: Rates of ammonium assimilation and nitrification in the bottom boundary layer (BBL), and denitrification in the permeable and non-permeable sediments of the Öre and Vistula estuaries in spring and summer. Rates are reported as the average and standard deviation. The number of replicates is shown in parentheses.**

| Site | Season | Ammonium assimilation BBL (nmol L⁻¹ d⁻¹) | Nitrification BBL (nmol L⁻¹ d⁻¹) | Denitrification Permeable sediment (μmol N m⁻² d⁻¹) | %Dn | Non-permeable sediment (μmol N m⁻² d⁻¹) | %Dn |
|---|---|---|---|---|---|---|---|
| Öre estuary[a] | Spring | 92 ± 70 (4) | 21 ± 7 (4) | - | - | n.d. | n.d. |
| | Summer | 218 ± 107 (7) | 49 ± 30 (7) | - | - | 138 ± 47 (65) | 93 |
| Vistula estuary[b] | Spring | 36 ± 16 (9) | 41 ± 22 (11) | 72 ± 38 (19) | 97 | 140 ± 52 (50) | 93 |
| | Summer | 319 ± 232 (10) | 64 ± 72 (7) | 354 ± 127 (49) | 97 | 349 ± 117 (21) | 90 |

Dn = coupled nitrification-denitrification, n.d. = not detectable.
[a] Nitrification rates including data from Happel et al. (2018); denitrification rates from Hellemann et al. (2017)
[b] Nitrification and ammonium assimilation rates from Bartl et al. (2018)



**Figures**

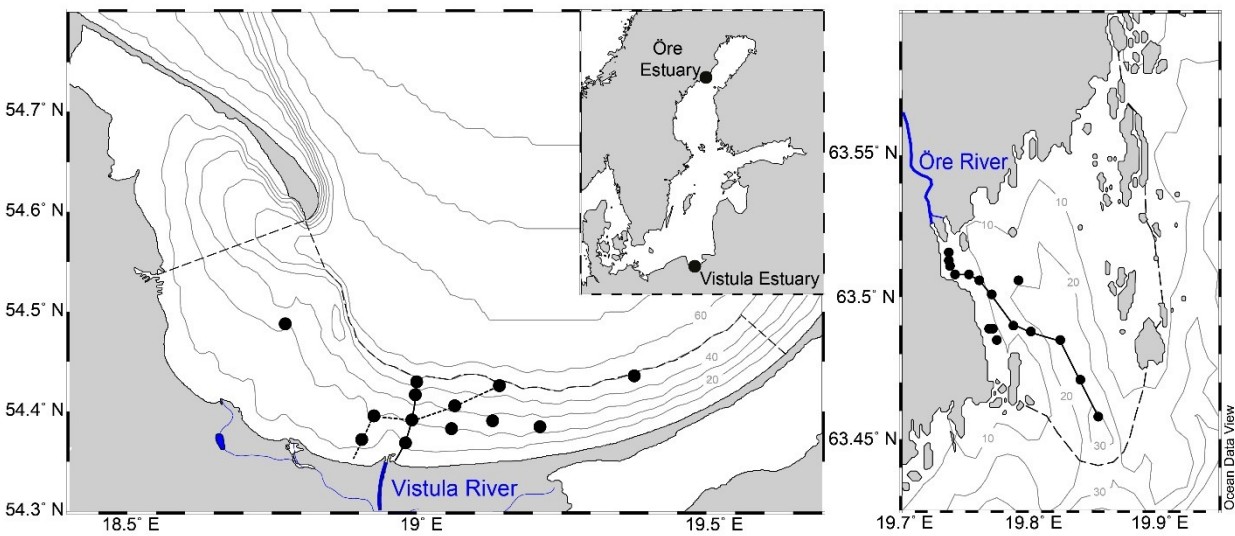

Figure 1: Map showing the locations of the Vistula estuary (left) and Öre estuary (right) in the Baltic Sea (inset). The boundaries of the estuaries are indicated by the dashed lines (see Section 2.1 for details). Lines along the station points represent the transects shown in Figures 2 and 3. Vistula estuary: VE I (solid line), VE II (dotted line).





**Figure 2: Environmental variables of the water column along a transect from the river mouth to the outermost station of the Öre (left) and Vistula (right) estuaries in spring. Bottom topography was estimated from the water depths of the stations. The dashed line represents the vertical extent of the BBL. The plots were derived from 12 (Öre estuary) and 4 (Vistula estuary) profiles using DIVA-gridding in Ocean Data view (Schlitzer, 2015).**





**Figure 3: Environmental variables of the water column along a transect from the river mouth to the outermost station in the Öre (left) and Vistula (right) estuaries in summer. Bottom topography was estimated from the water depths of the stations. The dashed line represents the vertical extent of the BBL. The plots were derived from 12 (Öre estuary) and 6 (Vistula estuary) profiles using DIVA-gridding in Ocean Data view (Schlitzer, 2015).**





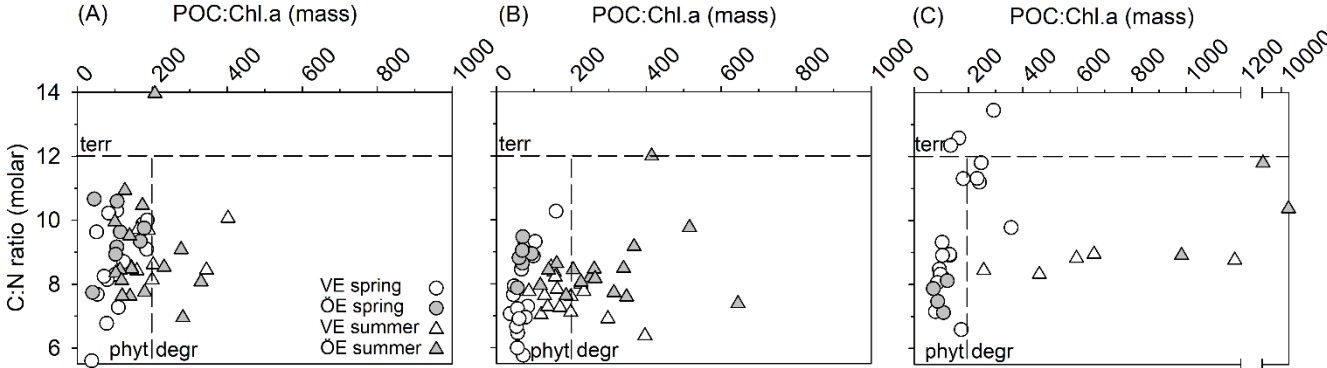

**Figure 4: Particulate C:N ratios plotted against POC:Chl.*a* ratios from the surface water (A), intermediate water depths (B) and bottom boundary layer (BBL, C) of the Vistula and Öre estuaries in spring and summer. Data at intermediate water depths are water depths of 10 m and 20 m in the Vistula estuary, and 5 m and 10 m in the Öre estuary. C:N ratios: terrestrial POM (terr) > 12 according to Savoye et al. (2003); POC:Chl.*a* ratios: newly produced phytoplankton POM (phyt) < 200 < degraded phytoplankton POM (degr) according to Cifuentes et al. (1988). Note the different scales of the POC:Chl.*a* ratios in panel C.**

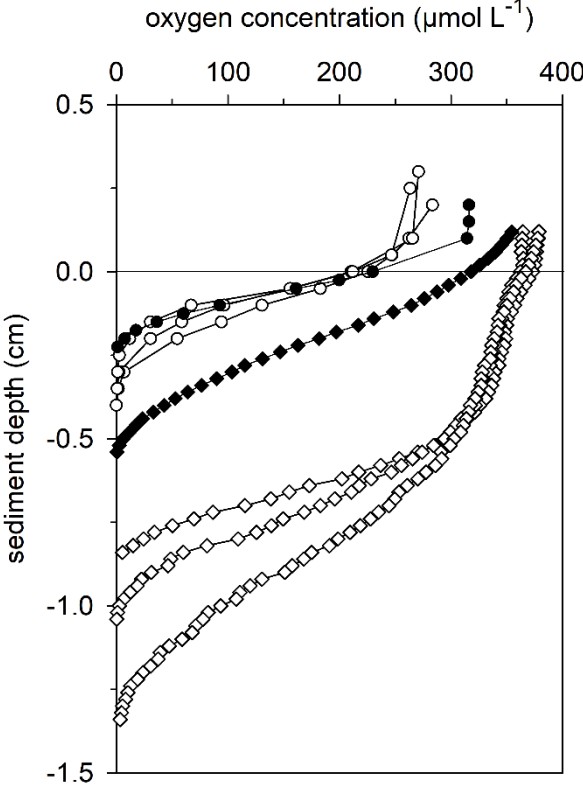

**Figure 5: Example pore-water oxygen concentration profiles in the permeable (white) and non-permeable (black) sediments of three representative stations in the Vistula estuary in spring (diamonds) and summer (circles). The zero line indicates the sediment surface.**





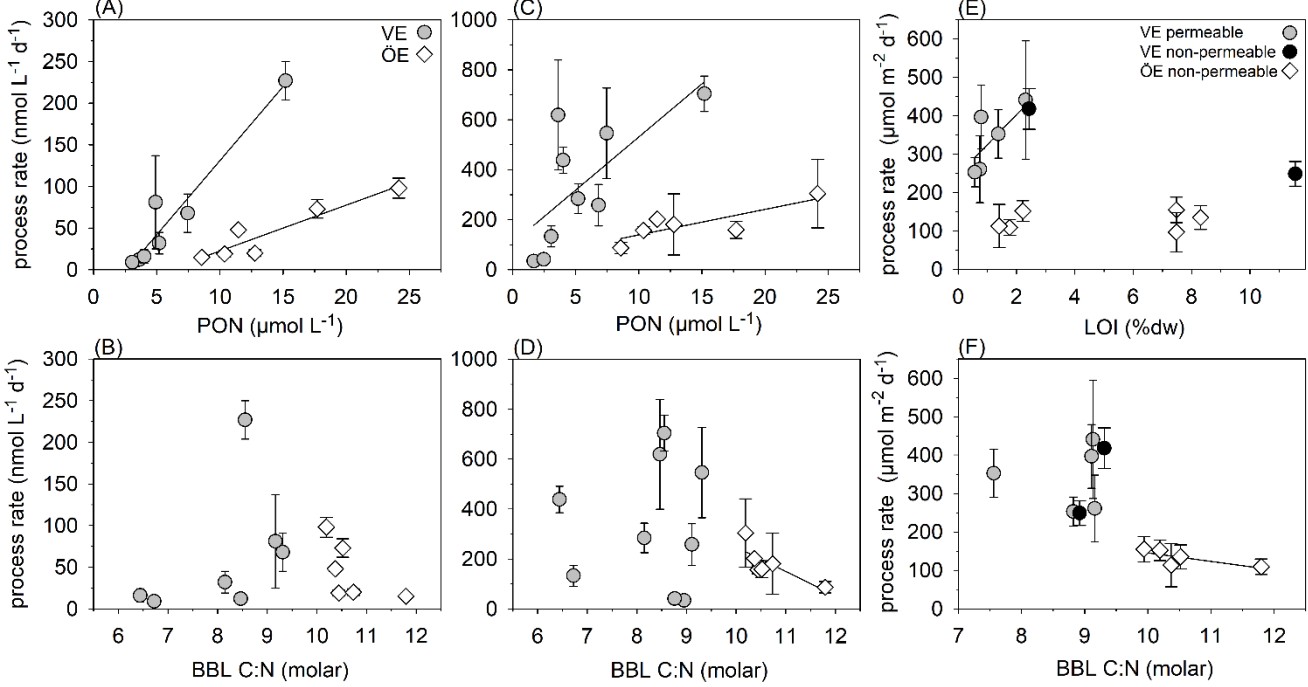

**Figure 6: Correlations of nitrification rates in the BBL with PON concentration (A) and particulate C:N ratio (B); ammonium assimilation rates in the BBL with PON concentration (C) and particulate C:N ratio (D); and coupled nitrification-denitrification rates in the sediment with LOI (E) and particulate C:N ratio (F). Solid lines represent significant correlations.**





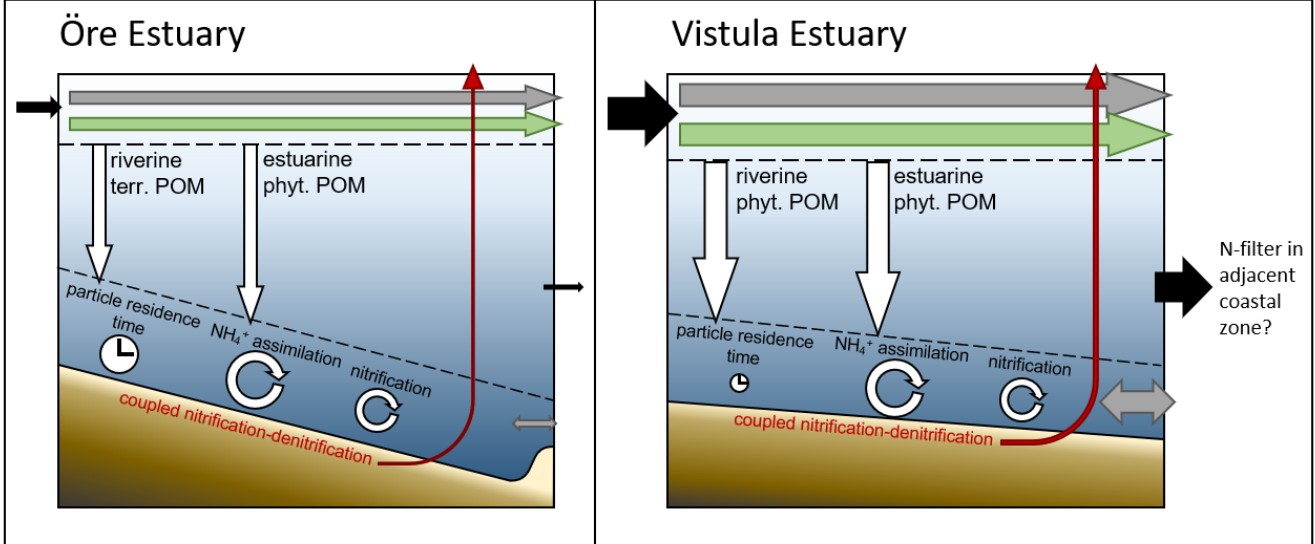

**Figure 7: Conceptual idea of the N-filter in the Öre (left) and Vistula (right) estuaries. Black arrows = estuarine inputs and outputs; gray arrows = transport; green arrows = uptake of riverine DIN into POM; white symbols = N-retention via sedimentation (downward arrows), nitrification and ammonium assimilation (circular arrows), and particle residence time (clock); red arrows =**
5 **N-removal via denitrification. In both estuaries, the uptake of riverine DIN via primary production leads to the sedimentation of easily degradable phytoplankton POM, which is mineralized to ammonium in the benthic system and subsequently retained in the BBL via ammonium assimilation and nitrification, or removed in the sediment via coupled nitrification-denitrification. In the Öre estuary, the limited transport of BBL water, and hence the long particle residence time, results in a high efficiency of the estuarine N-filter (Hellemann et al., 2017). In the Vistula estuary, the unrestricted bottom topography may lead to the enhanced transport of**
10 **DIN and estuarine phytoplankton-derived POM and a shorter particle residence time in the benthic system. This may limit estuarine N-filter efficiency via microbial N-retention and removal within the Vistula estuary. During the along-shore transport of DIN and POM, successive N-retention and removal in the adjacent coastal zone may further reduce outputs to the open sea.**