# Peer review of "Particulate organic matter controls benthic microbial N retention and N removal in contrasting estuaries of the Baltic Sea"

_Biogeosciences, 2018_

## Referee Comment (RC1) · Anonymous Referee #1 · 20 Nov 2018

**General Comments**

This paper compares two Baltic Sea estuaries that receive differing levels of riverine nitrate inputs. Using a variety of oceanographic measurements in the water column, the benthic boundary layer, and sediments, along with 15-N isotope labeling experiments, the authors compare differences in N cycling pathways between the two sites. In particular the authors examined the efficiency of each estuary as "filters" of land-derived N loading. The authors found that both estuaries had similar nitrification and assimilation rates but that denitrification rates were higher in the estuary that received higher N loading. Based on C:N and POC:Chl a ratios the authors conclude that the quality of organic matter plays an important role in controlling these N-cycling processes. They state that phytoplankton derived POM is an important link between riverine N loads benthic N-cycling by functioning as a temporary reservoir that increases the residence time of nitrogen in the estuary and allows more time for removal through denitrification. Generally I thought this was a well conducted study that sheds important light on the role of estuaries in removing land derived N, an important issue especially for the Baltic that suffers from N-loading induced eutrophication. I do however have a few comments that should be addressed in the final version of the manuscript.

**Specific Comments**

The authors should explicitly state how they define the BBL, since the BBL figures heavily in the analysis. They do mention it is based on density stratification and provide some references, but they should say exactly what criteria they used.

I find the presentation of the sediment characteristics in Table 3 a bit confusing. Why are the LOI values not provided as a mean and standard deviation like the OPD and NH4+. Also it seems that the range of LOI values is quite wide in some cases, (ie. Ore Spring 1.9-12.8 dw%) this implies to me there are many different sediment environments grouped together. Likewise the variability in the ammonium pools within sites is also quite large. It seems there might be more information here that could prove useful if the authors looked at this variability in greater detail. Presumably the high LOI values and high ammonium values come from the same sediment cores. Also I find the per area inventories of NH4+ a little bit of a strange way to present this, I think pore water ammonium profiles would useful to see as well. Why go through the effort of section cores and extracting porewater profiles with Rhizons if you are not going to show the changes with depth.

The authors mention that the NO3-+NO2-, PON, POC concentrations in the BBL were significant higher in Ore than in the Vistula estuary (again no statistics) they mention this is due to the long particle retention time of the Ore estuary compared to the more open unrestricted bottom topography of Vistula. However one of the most striking features of the BBL chemistry in Figure 3 and Table S1 is the

accumulation of NH4+ in the BBL of Vistula.  I am wondering how the authors reconcile these two points.

The authors make a big deal about the difference in bottom topography and the role it plays in differences in N-cycling, however the estimated bottom topography in Figures 2 and 3 does not look that different to me.  The authors should explain these differences in bottom topography in more detail to make a more convincing argument.  In section 2.1 the authors state that the "deep waters of the [Ore] estuary are confined by a small elevation (~30 m water depth) at its southern border." This to me implies there is some sort of sill that restricts the exchange of bottom water.  But I do not see any such feature in the map in Figure 1 or the bottom topography of Figures 2 or 3 that would restrict flow, 30m seems to be the deepest water depth and it appears to occur right at the estuary mouth.  The authors need to explain this a bit better, and provide stronger evidence for the restricted circulation.

In Section 2.2.1 the authors mention the high CDOM content of the Ores estuary and that they needed to do a correction to account for this in their nutrient analysis.  If this is the case I think it is likely that this CDOM would interfere with the in-situ Chl-a measurements using the optical sensor.  If the optical properties of the water in the both estuaries are different (due to CDOM levels) how accurate/comparable are the chlorophyll a cross-sections in Figures 2 and 3?

On line 385 the authors mention temperature as the factor determining higher ammonium assimilation rates in the summer, which could very likely be a contributing factor, but couldn't this also just be a concentration effect since NH4+ concentrations are so much higher in the summer (Figure 3).

Line 401 states:  " *In summary, the magnitude of nitrification and ammonium assimilation in the BBL was not influenced by the different trophic state or by seasonal differences. However, the regulation of those two processes differed depending on the trophic state, i.e. the availability of organic N from POM.*"  I do not understand this statement.  How is the magnitude of nitrification and ammonium assimilation not influenced by differences in trophic state, when figure 6 shows a clear correlation between these rates and the concentration of PON.

Line 429, The authors state the dominance of the NO3- source is controversial what is controversial about it? The authors should elaborate on this a bit more.

I think Figure 7 would be more effective if  numbers were put to the various arrows, it seems the authors have constrained at least some of these flows, and would be valuable to indicate which ones were known.

**Technical corrections**

Individual panels and figures 2 and 3 should be labeled.

Also is the horizontal axis for the right side of Figure 2 labeled correctly or should it be from 0-20km.

Line 259 "…In the BBL, particulate C:N ratios were significantly higher in the Öre (10.2 ± 0.9, n=9) than in the Vistula (8.6 ± 0.6, n=12) estuary in summer, but covered similar ranges in spring (ÖE: 7.1–10.8, n=7 and VE: 6.6–13.5, n=18)." – be consistent in how variability is reported 10.2+/- 0.9 vs 7.1-10.8.

---

## Author Comment (AC1) · 5 Dec 2018

**Response to RC1**

Dear Reviewer,
Thank you very much for your comments and suggestions. In the following you find the responses to your specific comments (bold) and changes in the manuscript text (italic).

Best regards,
Ines Bartl, Dana Hellemann and all co-authors

**The authors should explicitly state how they define the BBL, since the BBL figures heavily in the analysis. They do mention it is based on density stratification and provide some references, but they should say exactly what criteria they used.**

The reviewer is right and we will add more detailed information to the manuscript, section 2.1:

*The BBL is generally defined as the water layer directly above the sediment (Richards, 1990). It is characterized by turbulent boundary layer flow and mixing, which are typically fueled by bottom friction (Dade et al. 2001; Grant and Madsen, 1986; Thorpe, 2005). As turbulence and mixing lead to invariant values of potential density ($\sigma_\theta$) within the BBL (Turnewitsch and Graf, 2003), the vertical extent of the BBL can be determined based on the variation of the potential density ($\Delta\sigma_\theta$), i.e. the change of potential density over the change of depth. Thus the vertical extent of the BBL is defined by the lowermost point in the water column where the variation of the potential density exceeds a threshold of $\Delta\sigma_\theta < 0.01$ kg m$^{-3}$ (according Holtermann et al. 2012).*

This is the criteria that was chosen in our study (see Figure R1).

[Figure]

*Figure R1: Vertical profiles of the potential density $\sigma_\theta$ (black thick line) and the variation of the potential density $\Delta\sigma_\theta$ (white circles) and the threshold of $\Delta\sigma_\theta$ at 0.01 kg m$^{-3}$ (dashed line) of station VE05 from the Vistula estuary (A) and N3 from the Öre estuary (B). The difference between the bottom depth and the depth at which $\Delta\sigma_\theta$ exceeds the threshold makes the vertical extent of the BBL (grey box).*

**I find the presentation of the sediment characteristics in Table 3 a bit confusing. Why are the LOI values not provided as a mean and standard deviation like the OPD and NH4+. Also it seems that the range of LOI values is quite wide in some cases, (ie. Ore Spring 1.9-12.8 dw%) this implies to me there are many different sediment environments grouped together. Likewise the variability in the ammonium pools within sites is also quite large. It seems there might be more information here that could prove useful if the authors looked at this variability in greater detail. Presumably the high LOI values and high ammonium values come from the same sediment cores. Also I find the per area inventories of NH4+ a little bit of a strange way to present this, I think pore water ammonium profiles would useful to see as well. Why go through the effort of section cores and extracting porewater profiles with Rhizons if you are not going to show the changes with depth.**

The reviewer is correct in the assumption that in the Öre estuary two types of sediments were grouped together: silts and silty fine sands. While these are two distinct sediment types with different LOI values due to different organic matter sorption capacities (Mayer 1994a, Hedges & Keil 1995), they shared the same mass transport mechanism (diffusion and fauna-induced fluxes), as both had a too low permeability to enable advective pore-water flow. Thus, we grouped these two sediment types together as "non-permeable sediments". This was also supported by the similar oxygen penetration depth and denitrification rates of both sediment types (in detail explained in Hellemann et al. 2017). Probably only a small amount of the organic matter measured as LOI was labile, resulting in similar process rates despite different organic matter contents. As the different sediment types were discussed in detail in Hellemann et al. 2017, we did not want to repeat those results. We will change all values in Table 3 into mean and standard deviation.

The reviewer is also correct in the view, that some information is lost when calculating depth integrated element pools rather than showing actual pore-water profiles. However, our aim was to compare the $NH_4^+$ inventory in the sediments of the two estuaries, which give indications of long-term organic matter accumulation in the estuarine benthic system and the trophic condition in the estuary. Per area inventories of pore-water $NH_4^+$ have also been used to investigate different sediment types e.g. in the North Sea (Ehrenhauss et al. 2004).

Indeed, the $NH_4^+$ pools exhibit some variability within each estuary and sediment type ("permeable", "non-permeable"). This is due to the strong patchiness of sediment properties, well-known for coastal sediments. High LOI values co-occur with high pore-water $NH_4^+$ values only in the Öre estuary (Figure R2, grey symbols). However, similar to LOI, the different pore water $NH_4^+$ concentration did not influence rates of coupled nitrification-denitrification in the surface sediments there. In the Vistula estuary high LOI values do not always co-occur with high pore-water $NH_4^+$ values, likely due to high turnover of organic matter e.g. macrofaunal influence (Thoms et al. 2018).

We are of the opinion, that including the pore-water $NH_4^+$ profiles (Figure R3) in the manuscript would not add additional information in respect to the scope of our study. However, if the reviewer recommends adding such figures, we are happy to add them to the supplementary material.

[Figure]

*Figure R2: The pore-water NH₄⁺ pool (0-2cm) plotted against the organic matter content (LOI) in the surface sediment (0-2 cm).*

[Figure]

*Figure R3: Pore-water ammonium profiles of the Vistula (VE) and Öre estuary (ÖE) in spring and summer. Please note the different colour legends for the two estuaries. All sediments of the Öre estuary were non-permeable.*

**The authors mention that the NO3-+NO2-, PON, POC concentrations in the BBL were significant higher in Ore than in the Vistula estuary (again no statistics) they mention this is due to the long particle retention time of the Ore estuary compared to the more open unrestricted bottom topography of Vistula. However one of the most striking features of the BBL chemistry in Figure 3 and Table S1 is the accumulation of NH4+ in the BBL of Vistula. I am wondering how the authors reconcile these two points.**

The reviewer is right that the high summertime $NH_4^+$ concentrations in the BBL of the Vistula estuary contradicts the interpretation of the distribution of $NO_3^-+NO_2^-$, PON, and POC concentrations (line 302-310). We suggest that in the Vistula estuary thermohaline stratification reduced the vertical mixing and led to low bottom water flow-velocities (see sect. 4.1.3). In consequence, this led to low lateral transport and likely allowed the accumulation of $NH_4^+$ in the BBL in summer, despite the estuary´s open shape.

Changes from Line 306 onwards:
*The accumulation of elements in the benthic system of the Öre estuary is favored by the basin-like bottom topography and restricted bottom-water exchange, which allows a long particle residence time of more than one year (Brydsten and Jansson, 1989). Not only the coastal bottom topography, but also lateral bottom water flow or vertical mixing could influence the accumulation of dissolved nutrients or POM. The open shape and unrestricted bottom topography of the Vistula estuary may not allow accumulation in the benthic system. However, thermohaline stratification reduced the vertical mixing, and led to low bottom water flow-velocities (see sect. 4.1.3). The resulting low lateral transport likely allowed the accumulation of $NH_4^+$ in the BBL in summer, despite the open shape of Vistula estuary.*

Statistics were made as described in section 2.4 (lines 230-235). We decided to define the significance level in this section rather than adding it after every comparison/sentence in the results section, as we thought this would disturb the reading flow. If the reviewer recommends to change this, we are happy to do so.

**The authors make a big deal about the difference in bottom topography and the role it plays in differences in N-cycling, however the estimated bottom topography in Figures 2 and 3 does not look that different to me. The authors should explain these differences in bottom topography in more detail to make a more convincing argument. In section 2.1 the authors state that the "deep waters of the [Ore] estuary are confined by a small elevation (~30 m water depth) at its southern border." This to me implies there is some sort of sill that restricts the exchange of bottom water. But I do not see any such feature in the map in Figure 1 or the bottom topography of Figures 2 or 3 that would restrict flow, 30m seems to be the deepest water depth and it appears to occur right at the estuary mouth. The authors need to explain this a bit better, and provide stronger evidence for the restricted circulation.**

The bottom topography of the Öre estuary is well described in several studies (e.g. Brydsten et al., 1992: Brydsten and Jansson, 1989; Forsgren and Jansson, 1992; Malmgren and Brydsten, 1992) and as the reviewer recommended we will describe this in more detail (line 98):

*The Öre estuary is located on the Swedish coast of the Quark Strait, northern Baltic Sea (Fig. 1). It is partly separated from the open sea by an archipelago to the east and by land to the west,*

*and has a basic-like bottom topography with a hydrography depending on local wind conditions and river discharge (Brydsten, 1992, Fig. 1). The outlet of the Öre estuary in the south is relatively wide in the surface but becomes narrow at water depths >20m (Brydsten, 1992; Malmgren and Brydsten, 1992). A small elevation at ~30-25m depth separates the estuarine bottom waters from the open sea (Brydsten, 1992, Fig. 1). The Öre estuary covers an area of ~71 km$^2$ and has a volume of ~1 km$^3$ (SMHI, 2003). Inputs into the estuary originate from the Öre River, whose mean discharge is 36 m$^3$ s$^{-1}$, creating a river plume of 2–3 m vertical and ~10 km horizontal extent (Forsgren and Jansson, 1992). The water turnover time (estuarine volume/river discharge) is ~9 days (Engqvist, 1993). The sediments, covering ~20% of the estuarine are consist of silts, very fine and fine sands, all non-permeable (Hellemann et al., 2017).*

The sampling transect of the Öre estuary shown in Fig. 2 and 3 did not include the southern estuarine outlet (see Fig.1 in the manuscript), where the small elevation exists (compare to Fig. 1 in Brydsten 1992). We agree with the reviewer that Figures 2 and 3 should include this elevation to better visualize the differences in bottom topography between Öre and Vistula estuary. Example of transect including the elevation is shown in Figure R4.

[Figure]

*Figure R4: Updated transect from river mouth to estuary outlet of Öre estuary.*

**In Section 2.2.1 the authors mention the high CDOM content of the Ores estuary and that they needed to do a correction to account for this in their nutrient analysis. If this is the case I think it is likely that this CDOM would interfere with the in-situ Chla measurements using the optical sensor. If the optical properties of the water in the both estuaries are different (due to CDOM levels) how accurate/comparable are the chlorophyll a cross-sections in Figures 2 and 3?**

The optical properties of the water are indeed different in the two estuaries, with higher cDOM levels in the water of the Öre estuary. Figures 2 and 3 show Chl.a values which were measured with an optical sensor in the Öre estuary and manually with the fluorometric method in the Vistula estuary. The Chl.a-transects given in Figure 2 and 3 are only used to illustrate the presence or absence of phytoplankton in the water column and should not be compared to each other, which is now also explained in the figure captions (here as example for caption of Figure 2):

*Figure 2: Environmental variables of the water column along a sampling transect from the river mouth to the outlets of the Öre (left) and Vistula (right) estuaries in spring. Please note, due to different optical properties of the water and different measurement methods, the Chl. a*

*values are not directly comparable between the two estuaries, but provide qualitative information on the presence/absence of phytoplankton. Bottom topography was estimated from the water depths of the stations. The dashed line represents the vertical extent of the BBL. The plots were derived from 12 (Öre estuary) and 4 (Vistula estuary) profiles using DIVA-gridding in Ocean Data view (Schlitzer, 2015).*

For the calculation of the POC:Chl.a ratios in the Öre estuary, we used Chl.a concentrations manually measured with the HPLC analysis, kindly provided by Lumi Haraguchi (Aarhus University). Unfortunately, the manual Chl. a measurements were not done at all stations from the Öre estuary, so the data is sparser than for the Vistula estuary. We compared Chl. a concentrations from the optical sensors with the manually measured Chl.a concentrations and found the majority of values being similar (Figure R5). This shows that the optical Chl.a data from the Öre estuary given in Figure 2 and 3 are likely not strongly biased by cDOM.

[Figure]

*Figure R5: Chlorophyll a (Chl.a) concentrations measured by HPLC analysis vs. Chl.a concentrations measured with an optical sensor in the Öre estuary in spring (A) and summer (B).*

**On line 385 the authors mention temperature as the factor determining higher ammonium assimilation rates in the summer, which could very likely be a contributing factor, but couldn't this also just be a concentration effect since NH4+ concentrations are so much higher in the summer (Figure 3).**

The reviewer is correct in that the higher $NH_4^+$ concentrations in the BBL in summer than in spring may contribute to higher summertime $NH_4^+$ assimilation rates in both estuaries. In summer, i.e. at higher temperatures than in spring, degradation of organic matter to $NH_4^+$ is enhanced, and thus the combination of both environmental variables likely control seasonal variation of $NH_4^+$ assimilation rates. We changed the text at line 385 accordingly:

*Because ammonium assimilation is a substrate- and temperature-dependent process (Baer et al., 2014; Hoch and Kirchman, 1995), the combination of both, high $NH_4^+$ concentrations and elevated temperatures in summer enhanced ammonium assimilation rates in both estuaries.*

**Line 401 states: "*In summary, the magnitude of nitrification and ammonium assimilation in the BBL was not influenced by the different trophic state or by seasonal differences. However, the regulation of those two processes differed depending on the trophic state, i.e. the availability of organic N from POM.*" I do not understand this statement. How is the magnitude of nitrification and ammonium assimilation not influenced by differences in**

**trophic state, when figure 6 shows a clear correlation between these rates and the concentration of PON.**

We agree with the reviewer that this statement is not clearly formulated. We decided to remove this statement from the manuscript text.

**Line 429, The authors state the dominance of the NO3- source is controversial what is controversial about it? The authors should elaborate on this a bit more.**

This statement refers to the nitrate source of benthic denitrification in permeable sediments and is discussed in more detailed in line 429:

*In permeable sediments, the dominance of the $NO_3^-$ source is highly variable due to the complexity of pore-water flow (Kessler et al., 2013; Gihring et al., 2010; Marchant et al., 2016; Rao et al., 2007). On the one hand, pore-water flow was shown to stimulate nitrification by increasing the oxic sediment volume (Huettel et al. 1998, Giehring et al. 2010, Marchant et al. 2016), and to increase the areal oxic-anoxic interface across which $NO_3^-$ and $NH_4^+$ can be exchanged (Precht et al. 2004, Cook et al. 2006), thus favoring denitrification coupled to $NO_3^-$ produced in the sediment (Dn; Rao et al. 2008, Marchant et al. 2016). On the other hand, pore-water flow was also shown to separate the oxic inflow from the anoxic outflow zone, limiting the exchange of $NO_3^-$ and $NH_4^+$ within the sediment (Huettel et al. 1998, Cook et al. 2006, Kessler et al. 2012, 2013) and thus favoring denitrification of $NO_3^-$ from the near-bottom water (Dw; Cook et al. 2006, Kessler et al. 2012, 2013, Marchant et al. 2014).*

**I think Figure 7 would be more effective if numbers were put to the various arrows, it seems the authors have constrained at least some of these flows, and would be valuable to indicate which ones were known.**

We intended to present a conceptual view and thus had left out numbers in Figure 7. However, we agree, that it would be valuable to indicate what is known and what remains unknown, either by numbers or by question marks. Of the arrows presented in Figure 7, we measured the rates of ammonium assimilation, nitrification and denitrification. Transport rates are unknown, particle residence time is only estimated for the Öre estuary, and numbers for primary production and sedimentation rates may be found in the literature. We will revise Figure 7 and upload it latest in the revised manuscript.

---

## Referee Comment (RC2) · Anonymous Referee #2 · 14 Dec 2018

This study by Bartl et al. aims to address the role of two temperate estuaries with contrasting geomorphological, hydrological, biogeochemical features in processing riverine N and their efficiency as coastal filters for excessive land derived N in two different seasons. It is an honest attempt to emphasize the role of biological and microbial transformation processes which make these estuaries as traps for anthropogenic N. On a bird's-eye view, I see that the authors have tried to do a comparative analysis of various biogeochemical processes in Ore estuary and Vistula estuary in relation to their capacity to process anthropogenic N. Interestingly, the authors have used the available data on water column and sediment characteristics, and sedimentary denitrification rates already reported by Helleman et al. (2017) for Öre estuary but they have

additionally carried out ammonium assimilation and nitrification study in benthic boundary layer (BBL). Similarly, for Vistula estuary, the authors have used the available data on water column characteristics, and ammonium assimilation and nitrification rates in BBL already reported by Bartl et al. (2018; Estuarine coastal shelf science) but they have additionally carried out sediment characteristics and sedimentary denitrification study in order to address their central issue. Broadly, I see that the manuscript has not been written properly. It has a lot of unnecessary repetition of published stuffs by Helleman et al. (2017) and Bartl et al. (2018, ECSS). There are some gaps in terms of information and interpretation at certain sections and it has missed the focus at times. The major weakness of this manuscript is the discussion part which needs to be non-repetitive (in relation to Helleman et al. 2017 & Bartl et al.2018), novel on its own right and robust. Thus, the manuscript needs a lot of modification and re-interpretation at several sections to bring it to a decent publishable form. The study has the potential to contribute to our understanding of the role of estuaries as ecosystem service providers through the inter-connected biological-microbial-geochemical transformation processes. I would recommend it for publication, provided the authors make a major revision of the manuscript based on my critical observations as follows.

1. Line 34: A reference is needed for the last part of the sentence. 2. Line 85: Patsuszak et al. (2012) is used twice. Does this same reference give two different conclusions? If not, then two references can be given i.e. Patsuszak et al. (2012a) and Patsuszak et al. (2012b). 3. Section 2.1: This study emphasizes N transformation processes such as nitrification and denitrification in the benthic boundary layer which are sensitive to O2. But O2 regime of the estuarine water columns are not described. The authors should describe the variability of oxygen condition of two estuaries throughout year or at least from spring to summer based on previous study or this study. Line 121: .....to fill 30% (silt) to 50% (sand) of each one......What does this sentence mean? It sounds confusing. Please clarify by rephrasing the sentence. 4. Section 2.2.1: Was O2 measured in the water column? If yes, then how? By sensor coupled with CTD or measured analytically? Please give a brief description. The

authors should mention the thickness of BBL for both the estuaries in both spring and summer. Line 129-130: Water samples were..........If BBL thickness is just 20-40cm (If I understand correctly from Line 115) and sampler length is 0.5-1m. Then, how can you possibly say that the water sampler was completely inside BBL? Apparently, the sampler could also enclose the water above BBL. 5. Line 163: The authors mentioned that porewater was extracted at 2 cm interval from 5 cm to 11 cm depth by Rhizon tubings. But Seeburg-Elverfeldt et al. (2005) says that Rhizon tubings can extract porewater with a vertical resolution of 1 cm only. Please explain. 6. Section 2.3.1: It is not clear whether 100-170ml from BBL and 625 ml from water column were mixed together prior to $15NH4+$ enrichment or they were separately enriched with the substrate and incubated. If they were mixed, then what was the reason for that? BBL and the overlying water column can have different biogeochemical properties. So, if they were mixed and incubated with $15NH4+$, it cannot represent nitrification rates of BBL only and the aim of the study is to determine nitrification rate in benthic system not in water column. Please explain. O2 content of BBL and water column is not mentioned. Was O2 measured in sealed gas tight bags just prior to the experiment? Moreover, why were nitrification rates in the top oxic sediments not measured in both the estuaries and both seasons? The authors have emphasized the role of coupled nitrification-denitrification in these sediments. Then it makes sense to discuss benthic nitrification here which can have much higher rates compared to BBL nitrification due to higher availability of $NH4+$ diffusing from deeper sediments and its oxidation in top layer. 7. Section 2.3.2: The authors have not given a diagram for diffusive experimental set-up. 8. Line 194-200: For ÖE I and ÖE II, 4 replicates were made for each concentration but 12 replicates were made for 120$\mu$M VE I and 3 replicates for VE II. Why? Moreover, for ÖE I, ÖE II and VE I, three concentrations i.e. 40, 80, 120 $\mu$M 15NO3- were used but for VE II, four concentrations i.e. 30, 60, 90, 120 $\mu$M 15NO3- were used. Again, for permeable sediments of VE, three concentration treatments were given with 5-7 replicates. Why were 15NO3- concentrations different and why were no. of concentration treatments different? What was the rationale behind such varied

no. of treatments, replicates and concentrations between ÖE I, ÖE II, VE I and VE II? Why didn't the authors use same concentrations treatments and no. of replicates? For example, let's say, why couldn't they use 40, 80, 120$\mu$M 15NO3- treatments for all types of sediments with 4 replicates? 9. Line 204-205: Was the overlying water drawn only from the ports that were 5mm above oxic-anoxic interface or from all the ports lie above at 5mm resolution? 10. Line 212-213: What are the sampling time points? Was O2, NO3-, and NO2- measured in the overlying water at different time points? 11. Line 220-228: This paragraph needs to be rephrased. Risgaard-Petersen (2003) talks about the contribution of anammox to total N2 production from slurry incubation. But this study was based on intact core incubation. So first of all, please justify well that it can be applied to this study, given that the availability of 14NH4+ can be less in case of intact core incubation compared to slurry incubation which can affect p14 and p15 values described by Risgaard-Petersen et al (2003). Also I see that the first sentence of this paragraph i.e. from According to......till...1992) is a word to word copy from a sentence from Helleman et al (2017). This is not acceptable. Please rephrase the sentence. 12. Line 230: Replace it with significance of difference or variability. 13. Line 250 and Line 253: Both sentences contradict each other. Please rephrase the sentences. Sentence in line 250 means in both spring and summer POM in Öre river is dominated by terrestrial fraction but sentence in line 253 says in both spring and summer, POM is largely phytoplankton derived. 14. Line 273: It doesn't look so from the rates presented. 15. Line 273-276: These two sentences look contradictory. How can nitrification be positively correlated with POC if it shows negative trend with particulate C:N in case of Öre estuary? How can nitrification be positively correlated with PON, if it shows positive trend with particulate C:N? 16. Line 276-280: Same contradiction as in the case of nitrification. How can NH4+ assimilation be positively correlated with POC if it is negatively correlated with C:N? What is the logical explanation? 17. Section 3.2.2: The authors clearly concluded that there was no anammox and denitrification was the sole N loss process. What about DNRA? The authors didn't mention anything about it although it is only an N transformation process. I think the
authors are coming to conclusion here rather abruptly without considering findings of Jensen et al. (2011) in the Arabian Sea. Coupling of DNRA-Anammox can happen which can create an impression of denitrification signal and hence the conclusion can be misleading. Thus, the authors should relook at their incubation data and reinterpret if necessary. Again, the authors have not given any figure on 15N-labelled intact core incubation which is very important. Please present few figures depicting increase in 15N-N2O and 15N-N2 with time to support your conclusion on denitrification being a major N loss pathway. Similarly, if you find anammox and DNRA upon re-analysis of the incubation data, then please show the proof in terms of additional figures. Why coupled nitrification-denitrification was not correlated negatively particulate C:N in case of Vistula estuary? 18. Line 302-303: It doesn't look so. I don't see NO3-+NO2- in BBL of estuaries differing significantly if we strictly consider standard deviation (SD) given in Table S1. On the contrary, POC and PON in BBL of Vistula estuary are much higher than that in BBL of Öre estuary (Table 2). Please rephrase these sentences. 19. Line 306-309: This is true only for spring where we see high POC and PON in BBL of Öre compared to Vistula. But again on closer look, if we take SD and no. of replicates into account, POC in BBL of Öre is similar to that in Vistula and interestingly PON in BBL of Öre is higher than that in Vistula. This claim is anyway not true for summer. 20. Line 328: Delete Fig.4 from the sentence as it does not show C:N. 21. Line 330-332: Not a satisfactory explanation. Öre estuary has a sill and thus restricted exchange of estuarine water with seawater can likely cause more sedimentation within the estuary. 22. Line 333-334: C:N in Öre is higher than that in Vistula but POC:Chla in Vistula vary from 5.4 to 33.2 which is «200. How can it indicate degraded POM? Only because of C:N<12? 23. Line 336-338: Summer time POC:Chla in Ore varies from 12.6 to 140 that is <200. How can the POM be in degraded state? 24. Line 401 and Line 397: Please show the r and p values of the correlation. 25. Line 398: because the less-degraded POM in......This is questionable as POC:Chla is not above 200 rather «200. 26. Line 399: By contrast, the more degraded POM in........First of all, POC:Chla in both Öre and Vistula estuary are much lower than 200. So can we call

it degraded POM? Even though we assume higher POC:Chla (>200) as indicator for highly degraded POM, POM in Vistula estuary looks more degraded compared to that in Öre estuary. Not the other way. 27. Line 401: How significant is this correlation? What are the r and p values? 28. Line 402-404: These two sentences contradict each other. First sentence is questionable. I see a significant seasonal difference in the rates in both the estuaries. Please clarify the role of trophic state on these two processes. 29. Line 407-409: Difference in denitrification........How do you know that? Where is sedimentary Corg data and $\delta$13C-Corg for both the estuaries? 30. Line 415-419: While newly produced........If higher POM availability increased denitrification rates in sediments, then why not in water? Especially in BBL? 31. Line 421-423: what are the r and p values of the correlation? 32. Line: 425-427: Not written properly. OPD itself can get NO3- from BBL. Nitrification is significant in BBL and NO3- is not that low. 33. Line 427: Hence, only a small......How did the authors calculate that? Is there any nitrification rate measurement in the top oxic layer of the estuarine sediments? 34. Line 429: This sentence contradicts the previous sentence. If the dominant NO3- source is controversial, then how can you say that <10% of NO3- from BBL was removed by denitrification in permeable sediments of Vistula estuary? 35. Line 449: Please write it as During summer..... 36. Line 450-452: How is that possible? What about denitrification rate during spring? 37. Line 457-458: Despite their......What about spring? 38. Line 461-463: These two statements are contradictory. How would the authors reconcile these statements vis-a-vis their observation? 39. Line 465-468: What about DNRA? That would show that how much riverine N is preserved in estuarine sediments through DNRA. It is necessary to discuss that here. 40. 471-473: Through close......It is not necessary that only POM controlled benthic nitrification. What about benthic NH4+ efflux? 41. Line 482: What are the DNRA rates? 42. Line 490-492: We thus hypothesize........ How do the authors say it is a coast parallel transport? Is there any reference? The riverine flow may be perpendicular to the coast into the Baltic. 43. References: Holtermann et al. (2014), Risgaard-Petersen et al. (2004) and Schultz (2000) are not cited in the text. Schultz (2005) is missing in the

reference list. 44. Table 2: The authors need to show C:N in a column here. 45. Table 3: How have NH4+ surface pool and NH4+ deep pool been defined? Up to what depth you consider it as surface pool? Please mention clearly in the table caption. 46. Table 4: I don't see any denitrification rate in permeable sediments of Öre estuary. Was it not measured or it is not detectable? "–" symbol doesn't mean anything. Please clarify. 47. Figure 2 & 3: The PON plots for Öre estuary are reproduced from Helleman et al (2017). So please mention the reference clearly in the figure captions. 48. Figure 4: This figure contradicts the data in Table 2 and Table S1. If we calculate POC:Chla from Table 2 and S1, they range from 5.4 to 140. How come Fig.4 shows such higher POC:Chla values then? 49. Figure 5: Shows vertical O2 profile of Vistula estuary sediments. But what about that of Öre estuary sediments? The authors should show that also. 50. Figure S1: The authors should point out the ports through which water sample was collected. Please point out the water above the sediments. 51. Table S1: Looks a bit confusing and unexplained. River plume very much prevails within these two estuaries and occupies a depth range of up to 3m in case of Öre estuary and up to 12m in case of Vistula estuary. So when we say river plume here that actually means surface water of estuary. So, why can't the authors consider the depth from the river plume till bottom? If they do so, then I believe the so-called surface here would actually be a depth of 3m in case of Öre and 12m in case of Vistula. The authors should clear the confusion and mention terms in a logically correct way. Additionally, I believe a column for POC:Chla is necessary in this table. Overall comments & suggestions: I suggest the authors to be careful about not repeating the description of sampling methods, analysis/experiment methods and results which are already reported by Helleman et al. (2017) and Bartl et al. (2018) for these two estuaries. For example: Do not describe the water column sampling methods, sediment sampling methods, analysis methodology, denitrification experiment method and their results in details for Öre estuary because these are already published by Helleman et al. (2017). But you can retain everything about NH4+ assimilation and nitrification in Öre estuary. Similarly, for Vistula estuary, avoid detailed description of column sampling

methods, analysis methods and ammonium assimilation and nitrification experiment methods in BBL and their results because these are already published by Bartl et al. (2018). But you can retain everything about sediment sampling and analysis methods, denitrification experiment methods and their results. However, the authors can use the published data and their own generated data for the discussion since it's a comparative account study. The authors have not measured DNRA rates and have not discussed its role in transforming riverine N to NH4+ in the estuarine sediments. They have not also measured sedimentary nitrification rates which is very important. I did not see any discussion on benthic N (NO3- uptake or NH4+ release) exchange. All these could have made the discussion on benthic N cycling robust. However, the authors should use the published data (if any) on benthic nitrification, benthic DNRA and benthic N exchange and thoroughly discuss the interplay of all N cycling processes in relation to net N loss/ immobilization in these sediments in the discussion section in general and section 4.2.4 and section 4.3 in particular. I suggest the authors to relook into the classic integrated discussion on benthic N cycling in the Gulf of Bothnia by Bonaglia et al. (2017). In order to show the efficiency of these two estuaries as coastal filters, the authors should mention how much % of riverine N is ultimately lost in estuarine sediments through denitrification and/or anammox (if any), how much % is immobilized in sediments through DNRA and how much % is transported out of estuary to the coastal sea. Overall, I would suggest the authors to revise the manuscript by showing novelty of their study objectives, approach and findings which would make it appear as different from studies by Helleman et al. (2017) and Bartl et al. (2018).

Please also note the supplement to this comment:
https://www.biogeosciences-discuss.net/bg-2018-450/bg-2018-450-RC2-supplement.pdf

---

## Author Comment (AC2) · 16 Jan 2019

Review comments on "Particulate organic matter controls benthic microbial N retention and N removal in contrasting estuaries of Baltic Sea" by Bartl et al.

Dear Reviewer, Thank you very much for your valuable and detailed comments and suggestions. In the following you find the responses to your comments and the changes we will apply to the manuscript (italic).

1. Line 34: A reference is needed for the last part of the sentence.

Changes (underlined) from line 34 ff: Human nitrogen (N) utilization, especially in agriculture (Galloway and Cowling, 2002; Rabalais, 2002) has strongly increased riverine N inputs into coastal zones (Howarth et al. 1996), resulting in the eutrophication of coastal waters (Nixon 1995, Howarth & Marino 2006).

2. Line 85: Patsuszak et al. (2012) is used twice. Does this same reference give two different conclusions? If not, then two references can be given i.e. Patsuszak et al. (2012a) and Patsuszak et al. (2012b).

We apologize for this mistake. The first reference in line 85 will be deleted.

3. Section 2.1: This study emphasizes N transformation processes such as nitrification and denitrification in the benthic boundary layer which are sensitive to O2. But O2 regime of the estuarine water columns are not described. The authors should describe the variability of oxygen condition of two estuaries throughout year or at least from spring to summer based on previous study or this study.

The oxygen concentrations are indeed important and have been reported in this study in lines 241-242. We emphasize them more in the study area description and report the changes to the text below: line 100 ff: The water column of the Öre estuary is oxic throughout the year, ranging from ∼250 $\mu$mol L-1 in summer to ∼450 $\mu$mol L-1 in spring (monitoring data from dBotnia 2016; SMHI 2016). line 106 ff: The water column of the Vistula estuary is oxic throughout the year with small seasonal differences, i.e. lower oxygen concentrations in summer than in winter or spring (Bartl et al., 2018). Under specific conditions (floods, high organic matter mineralization under stagnant stratification) coastal hypoxia has been observed in the Vistula estuary (Conley et al., 2011; SMHI 2011). However, these were not met during our study.

3.1 Line 121: .....to fill 30% (silt) to 50% (sand) of each one......What does this sentence mean? It sounds confusing. Please clarify by rephrasing the sentence.

The reviewer is correct that the percentage plus sediment type leads to confusion. Changes from line 121 ff: Subsamples for denitrification rate measurements (n = 12

per site, except VE I: n = 20) and pore-water oxygen profiles (n = 3 per station) were collected in acrylic cores (iØ 2.3 cm, length 20 cm, except VE I: 15 cm). These were pushed gently into the sediment so that they were filled to 30% of volume with sediment in silty sediments, and to 50% of volume in sandy sediments. The remaining volume was overlying water. The cores were then closed without gas headspace.

4. Section 2.2.1: Was O2 measured in the water column? If yes, then how? By sensor coupled with CTD or measured analytically? Please give a brief description.

Changes from line 127 ff: In the Vistula estuary, water column measurements were carried out with a Seabird CTD-system (Seabird 911plus, Seabird Scientific) equipped with sensors for the measurement of the dissolved oxygen concentration (SBE43, Seabird-Scientific) (Bartl et al., 2018). In the Öre estuary, water column measurements were carried out by a Seabird CTD (SBE19plus, ÖE I; SBE19plus V2, ÖE II; Seabird Scientific), and oxygen was measured by an optode (4330, Aanderaa) attached to a Seaguard-CTD (Aanderaa). Oxygen concentrations from the sediment overlying core water were determined via Winkler-titration (Grasshoff et al., 1983; Winkler, 1888).

4.1 The authors should mention the thickness of BBL for both the estuaries in both spring and summer. Line 129-130: Water samples were..........If BBL thickness is just 20-40cm (If I understand correctly from Line 115) and sampler length is 0.5-1m. Then, how can you possibly say that the water sampler was completely inside BBL? Apparently, the sampler could also enclose the water above BBL.

Reviewer#1 also commented on the BBL thickness and we will describe the determination of the BBL thickness in more detail. The vertical extent of the BBL is given in Table S1. Changes line 128 ff: The BBL is defined as the water layer directly above the sediment (Richards, 1990) and is characterized by high turbulence and mixing, which are typically fueled by bottom friction (Dade et al. 2001; Grant and Madsen, 1986; Thorpe, 2005). As turbulence and mixing lead to invariant values of potential density (ïĄş$\theta$) within the BBL (Turnewitsch and Graf, 2003), the vertical extent of the BBL can

be determined based on the variation of the potential density ($\Delta$ïĄş$\theta$), i.e. the change of potential density over the change of depth. Thus the vertical extent of the BBL is defined by the lowermost point in the water column where the variation of the potential density exceeds a threshold of $\Delta$ïĄş$\theta$ < 0.01 kg m-3 (according Holtermann et al. 2012). Sediment overlying water taken from sediment cores (20 – 40 cm) were always within the BBL. Bottom water samples taken with Niskin bottles (0.5 – 1 m length) could only be considered as BBL samples in 57% of all sampled stations, because the vertical extent of the BBL in Vistula and Öre estuary ranged between 1 m and 7 m (Table S1).

5. Line 163: The authors mentioned that porewater was extracted at 2 cm interval from 5 cm to 11 cm depth by Rhizon tubings. But Seeburg-Elverfeldt et al. (2005) says that Rhizon tubings can extract porewater with a vertical resolution of 1 cm only. Please explain.

Seeberg-Everfeldt et al. (2005) recommend a vertical resolution of 1 cm as highest possible resolution when sampling pore-water with rhizons. This means an interval of < 1 cm should not be applied because then the pore-water catchment area of the single sampling depths would overlap and thus bias pore-water nutrient concentrations. However, an interval of > 1 cm is not problematic. At sediment depths > 5 cm, ammonium concentrations generally show a clear increasing trend in coastal Baltic sands and muds (Bonaglia et al., 2014; Lipka et al., 2018; Lenstra et al., 2018; Thoms et al., 2018) which can be well captured at a resolution of 2 cm intervals.

6. Section 2.3.1: It is not clear whether 100-170ml from BBL and 625 ml from water column were mixed together prior to 15NH4 + enrichment or they were separately enriched with the substrate and incubated. If they were mixed, then what was the reason for that? BBL and the overlying water column can have different biogeochemical properties. So, if they were mixed and incubated with 15NH4+, it cannot represent nitrification rates of BBL only and the aim of the study is to determine nitrification rate in benthic system not in water column. Please explain.

We agree with the reviewer that the method description in section 2.3.1 is misleading and must be clarified. No water samples were mixed, they were separately enriched with the tracer. Changes line 173ff: Water samples for incubations with 15N-NH4+ tracer (Damashek et al., 2016; Ward, 2005) were collected from the bottom water (water sampler/Niskin bottle) and from the sediment overlying core water and processed as described in detail by Bartl et al. (2018). Briefly, six polycarbonate bottles (sediment overlying water: 100mL bottle volume, VE II, 170mL bottle volume, VE I, ÖE I, II; Niskin bottle: 625 mL bottle volume, all field campaigns) were filled with water and sealed gas-tight.

6.1 O2 content of BBL and water column is not mentioned. Was O2 measured in sealed gas tight bags just prior to the experiment?

Oxygen concentrations are given in Table S1 and described in lines 241-242. Oxygen was not measured in sealed gas tight bags prior to the experiment. We did not use gas tight bags for incubations, but polycarbonate bottles. Please see answer to comment no. 4 regarding oxygen measurements.

6.2 Moreover, why were nitrification rates in the top oxic sediments not measured in both the estuaries and both seasons? The authors have emphasized the role of coupled nitrification-denitrification in these sediments. Then it makes sense to discuss benthic nitrification here which can have much higher rates compared to BBL nitrification due to higher availability of NH4 + diffusing from deeper sediments and its oxidation in top layer.

Indeed, nitrification rates were not determined in the surface sediments which is unfortunate. However, the IPT gives denitrification rates based on nitrate from the sediment overlying BBL water and based on nitrate from nitrification in the sediments (see lines 226-228; 287-288; 425-431). For your information, we estimated nitrification rates in the sediment from the Vistula estuary in spring, based on the sum of coupled nitrification-denitrification (Dn) and total nitrate fluxes out of the sediment (according
to Bonaglia et al., 2014). Since no comparison with the Öre estuary was possible we decided to leave these data out. We explain below how our estimates of nitrification in sediments compare to other sites, where rates have been measured. However, we do not want to add the text to the manuscript because it is a bit beyond the scope of our study. Nitrification rates in the sediment of the Vistula estuary were estimated from the sum of Dn (this study) and total nitrate fluxes from the sediment to the overlying BBL water (in situ incubations with chamber lander, Thoms et al., 2018). In the permeable sediment, mean Dn is 58 $\mu$mol m-2 d-1 and the mean nitrate flux is 507 $\mu$mol m-2 d-1 (n=3); resulting in a nitrification rate of 565 $\mu$mol m-2 d-1. In the non-permeable sediment, mean Dn is 110 $\mu$mol m-2 d-1 and the nitrate flux is 140 $\mu$mol m-2 d-1 (n=1); resulting in a nitrification rate of 250 $\mu$mol m-2 d-1. These estimates fall in line with other estimates from muddy sediments of the Baltic estuary Himmerfjärden ($\sim$389 $\mu$mol m-2 d-1; Bonaglia et al., 2014). Compared to rate measurements, these nitrification rates in the permeable sediment are higher than wintertime nitrification rates in subtidal North Sea sediments (very fine sand: 198 $\mu$mol m-2 d-1, fine sand: 216 $\mu$mol m-2 d-1; Lohse et al., 1993), and higher than springtime nitrification rates in intertidal sands of the North Sea (342 $\mu$mol m-2 d-1; Jensen et al., 1996). The estimated nitrification rate in the non-permeable sediment of the Vistula Estuary is similar to measured springtime nitrification rates in muddy sediments of the Baltic Gulf of Finland (286 $\mu$mol m-2 d-1; Jäntii et al., 2011). Comparing the estimates of nitrification rates in the sediment to areal nitrification rates of 131 $\mu$mol m-2 d-1 measured in the BBL of the Vistula estuary in spring (integrated over the vertical BBL extent of 3.2 m), nitrification rates in the sediment are 2 – 4 times higher than in the BBL, most likely due to the higher availability of NH4+ in the sediment (sediment at 1 cm/2 cm depth: 6.1/14.8 $\mu$mol L-1; BBL: 0.6 $\mu$mol L-1).

7. Section 2.3.2: The authors have not given a diagram for diffusive experimental set-up.

Diffusive core incubations are an established and widely used incubation method for

cohesive sediments e.g. Jørgensen & Sørensen 1985, Nielsen 1992, Nielsen & Glud 1996, Sundbäck et al. 2006, Hietanen & Kuparinen 2008, Jäntti et al. 2011, Bonaglia et al. 2014, Bonaglia et al. 2017. To reduce the number of figures in this paper we decided to explain the diffusive design in the text (line 192-196 of the manuscript) and only show an illustration of the new advective incubation set-up, which has been designed for this study and needs detailed explanation. Nevertheless, if the reviewer feels that an illustration of the diffusive set-up is necessary, we will add one in the supplements.

8. Line 194-200: For ÖE I and ÖE II, 4 replicates were made for each concentration but 12 replicates were made for $120\mu$M VE I and 3 replicates for VE II. Why? Moreover, for ÖE I, ÖE II and VE I, three concentrations i.e. 40, 80, 120 $\mu$M 15NO3- were used but for VE II, four concentrations i.e. 30, 60, 90, 120 $\mu$M 15NO3- were used. Again, for permeable sediments of VE, three concentration treatments were given with 5-7 replicates. Why were 15NO3- concentrations different and why were no. of concentration treatments different? What was the rationale behind such varied no. of treatments, replicates and concentrations between ÖE I, ÖE II, VE I and VE II? Why didn't the authors use same concentrations treatments and no. of replicates? For example, let's say, why couldn't they use 40, 80, 120$\mu$M 15NO3- treatments for all types of sediments with 4 replicates?

The sampling campaigns in the Vistula and the Öre estuary have been carried out over a period of two years, during which improvements in the incubation design were undertaken, such as increasing the number of replicates (from three to four) in favor of a decrease in the number of used 15N-NO3- concentrations (from four to three). These changes do not affect the resulting data: the concentration series is used to check whether the requirements of IPT (homogeneous distribution of the tracer and nitrate limitation of the sediment, Nielsen 1992) are fulfilled (plotting D15 against increasing 15N-NO3- concentrations), as well as to check for a contribution of anammox to total N2 production (plotting D14 against 15N-NO3- concentrations, Risgaard-Peterson et

al. 2003, see also response to comment no. 11). These tests are done by regression analysis. In order to ensure adequate number of replicates for the regression analysis we decided to use more replicates per concentration, with fewer concentrations, as samples were sometimes lost, unrepresentative or disturbed. In case of VE I (spring sampling), 12 replicates were run at the concentration 120 $\mu$M, to measure labelled N2 production over time, which had earlier been shown to be insignificant due to seasonal limitation of denitrification activity in spring (explained in section 4.2.2); it was thus used as an internal test.

9. Line 204-205: Was the overlying water drawn only from the ports that were 5mm above oxic-anoxic interface or from all the ports lie above at 5mm resolution?

Water was only drawn from the one port that was located $\sim$5mm above the approximated oxic-anoxic interface and recirculated during the incubation time.

10. Line 212-213: What are the sampling time points? Was O2, NO3-, and NO2- measured in the overlying water at different time points?

Time points were start (0 h) and end (3-5 h) for N2 and start (0 h) only for overlying water O2 NO3- and NO2- concentrations. We incubated samples in a concentration series, not a time-series. All cores had a total incubation time of 3-5h (line 196, 211) without sampling in between (neither for labelled N2 production, nor O2, NO3- or NO2-).

11. Line 220-228: This paragraph needs to be rephrased. Risgaard-Petersen (2003) talks about the contribution of anammox to total N2 production from slurry incubation. But this study was based on intact core incubation. So first of all, please justify well that it can be applied to this study, given that the availability of 14NH4+ can be less in case of intact core incubation compared to slurry incubation which can affect p14 and p15 values described by Risgaard-Petersen et al (2003).

According to Risgaard-Peterson et al. (2003), core-samples incubated in a concentra-

tion series can be used as an alternative to slurry incubations to indicate a contribution of anammox (page 72, first paragraph in Risgaard-Peterson et al., 2003). Following the method, a contribution of anammox to total N2 production is indicated, when the production of 14N-N2 correlates positively with the concentrations of added 15NO3-tracer. In such case, calculations have to be performed to distinguish N2 production from anammox and denitrification. In this study, 14N-N2 never correlated with the added tracer concentrations, indicating no contribution of anammox to total N2 production, which leaves denitrification as the sole N2 production process.

11.1 Also I see that the first sentence of this paragraph i.e. from According to......till...1992) is a word to word copy from a sentence from Helleman et al (2017). This is not acceptable. Please rephrase the sentence.

We rephrased the sentence so that it now reads: A contribution of anammox to the measured N2 production is indicated, when the production rate of 14N-N2 (D14, calculated with the IPT, Nielsen, 1992) correlates positively with the increasing 15N-NO3$-$ concentration in the incubation. In this case, calculation of N2 production needs to distinguish between denitrification and anammox rates, following Risgaard-Petersen et al. (2003).

12. Line 230: Replace it with significance of difference or variability.

Changes from line 230: Significance of differences between the factors 'site' (Öre estuary, Vistula estuary), 'season' (spring, summer) and 'sediment type' (permeable, non-permeable) was tested using. . .

13. Line 250 and Line 253: Both sentences contradict each other. Please rephrase the sentences. Sentence in line 250 means in both spring and summer POM in Öre River is dominated by terrestrial fraction but sentence in line 253 says in both spring and summer, POM is largely phytoplankton derived.

Changes from line 250 ff: In the Öre River and river plume, POM contained a large

share of terrestrial POM in both seasons, while the Vistula River and river plume were dominated by phytoplankton-derived POM (Table 2). The terrestrial origin of POM from the Öre River and river plume was reflected by the high C:N ratios and low $\delta$13C-POC values, neither of which occurred in the BBL of the Öre estuary or in the Vistula River and estuary (Table 2). In the estuarine water column (river and river plume excluded), the POM contained a large share of phytoplankton-derived POM in both estuaries and in both seasons (Table 2). This was further reflected in the high Chl.a concentrations measured throughout the water column in spring and in the surface water in summer (Fig. 2, 3).

14. Line 273: It doesn't look so from the rates presented.

Also reviewer#1 commented on the missing results from the statistical analyses. We defined the significance level in section 2.4 (line 235) rather than adding it after every comparison/sentence in the results section, as we thought this would disturb the reading flow. However, we see that there is a need to add the statistical results in the text, which we will do in the revised manuscript. In this case (line 273) the significance level of the Kruskal-Wallis-Test is p=0.478 and clearly shows that there are no significant differences between nitrification rates. We added the Box-Whisker-plot here, to visualize this result (Figure R1).

Changes at line 273 ff: Nitrification rates in the BBL did not significantly differ either between seasons or between estuaries (KW-Test, p=0.478; Table 4).

Figure R1: Nitrification rates in the BBL of the Öre and Vistula Estuary in spring (white) and summer (grey).

15. Line 273-276: These two sentences look contradictory. How can nitrification be positively correlated with POC if it shows negative trend with particulate C:N in case of Öre estuary? How can nitrification be positively correlated with PON, if it shows positive trend with particulate C:N? 16. Line 276-280: Same contradiction as in the case of nitrification. How can NH4+ assimilation be positively correlated with POC if it

is negatively correlated with C:N? What is the logical explanation?

Comments 15 and 16 are answered together in the following: In both estuaries, nitrification rates and ammonium assimilation rates are positively correlated with both PON and POC concentrations (see lines 274-275 and Figure 6 A, C). In the Öre estuary, nitrification rates show a negative trend with the C:N ratios (Kendall's $\tau$= −0.52, p=0.10, n=7; Figure 6 B) and ammonium assimilation rates show a significant negative correlation with the C:N ratio (Kendall's $\tau$= −0.71, p=0.02, n=7, Figure 6 D). A positive correlation between a rate and the POC or PON concentration does not necessarily imply that there should also be a positive correlation with the ratio of POC:PON. In the case of the Öre estuary, the C:N ratio is negatively correlated to PON and POC concentrations (Kendall's $\tau$= −0.62, p=0.05, n=7; Figure R3). Consequently, nitrification and ammonium assimilation rates seem to be influenced by a combination of the concentration and the ratio of POC and PON. Interestingly, lowest C:N ratios were found at greatest depth (Kendall's $\tau$= −0.81, p=0.01, n=7), which indicates accumulation of phytoplankton-derived POM in the deeper parts of the Öre estuary. Unfortunately, there was a mistake in the plotted C:N ratios of the Vistula estuary in panels B and D of Figure 6. We apologize for this and added the corrected figure below (Figure R2). We would not interpret the relationship of nitrification or ammonium assimilation rates with the C:N ratio in the Vistula estuary as positive trend, which is underlined by the lacking correlations (nitrification vs C:N: Kendall's $\tau$= −0.04, p=0.93, n=7; ammonium assimilation vs C:N: Kendall's $\tau$= −0.03, p=0.94, n=9).

Figure R2: Correlations of nitrification rates in the BBL with PON concentration (A) and particulate C:N ratio (B); ammonium assimilation rates in the BBL with PON concentration (C) and particulate C:N ratio (D); and coupled nitrification-denitrification rates in the sediment with LOI (E) and particulate C:N ratio (F). Solid lines represent significant correlations. Please note the different scaling of C:N ratios compared to figure 6 in the manuscript.

Figure R3: Correlations of PON (left panel) and POC (right panel) concentration with

the particulate C:N ratio.

17. Section 3.2.2: The authors clearly concluded that there was no anammox and denitrification was the sole N loss process. What about DNRA? The authors didn't mention anything about it although it is only an N transformation process. I think the authors are coming to conclusion here rather abruptly without considering findings of Jensen et al. (2011) in the Arabian Sea. Coupling of DNRA-Anammox can happen which can create an impression of denitrification signal and hence the conclusion can be misleading. Thus, the authors should relook at their incubation data and reinterpret if necessary.

We did not measure DNRA rates in this study and no significant anammox rates were found, as analyzed with a concentration series following Risgaard-Petersen et al (2003; see answer to comment no. 11, manuscript line 220-226). In case DNRA was active in the estuarine sediments, it could have produced 15N-NH4+ based on transformation of 15N-NO2- originating from the reduction of the added 15N-NO3- tracer (during IPT incubation). However, only a further combination of this 15N-NH4+ with 14N-NO2- or 15N-NO2-, such as described by Jensen et al. (2011), could result in additional single (29N2) or double labeled N2 (30N2) that would not have originated from denitrification and thus would violate the binomial distribution required for denitrification calculations based on IPT (Nielsen 1992). Without anammox, the 15N-NH4+ produced by DNRA would simply stay within the sediment or be nitrified again to 15N-NO3-, but not interfere with the production of N2 via denitrification. Consequently, as we did not find any sign of anammox, we are certain that the measured single and double labeled N2 production came from denitrification only.

17.1 Again, the authors have not given any figure on 15N-labelled intact core incubation which is very important.

Please see response to comment no. 7 regarding the graphic display of core-incubations.

17.2 Please present few figures depicting increase in 15N-N2O and 15N-N2 with time to support your conclusion on denitrification being a major N loss pathway. Similarly, if you find anammox and DNRA upon re-analysis of the incubation data, then please show the proof in terms of additional figures.

The presense / absence of anammox, thus its significant /non-significant contribution to total N2 production and the consequential role of denitrification in N2 production were investigated by concentration series (Risgaard-Petersen et al. 2003), not in time-series. In the concentration series, D15 (= the denitrification of 15N-NO3-) has to correlate with increasing tracer concentration to fulfill basic requirements of IPT (homogeneous distribution of the tracer and nitrate limitation of the sediment, i.e. basically homogeneous uptake of the tracer, Nielsen 1992), whereas D14 (= the true denitrification) should be independent of tracer concentration, if no anammox occurs. In contrast, a significant increase of D14 with increasing tracer concentration would indicate anammox, for which then separate calculations need to be applied, following Risgaard-Petersen et al. (2003). These relations were tested with regression analyses (significance level $p < 0.05$). Below an example plot of N2 data without contribution of anammox (i.e. D14 not dependent on increasing tracer concentration: A= Öre Estuary, station N34, summer; B= Vistula Estuary, station VE05, summer), as was the case in all incubations.

Figure R4: Denitrification of labeled 15N-NO3- (D15) and unlabeled 14N-NO3- (D14) at increasing 15N-NO3- tracer concentration for the Öre estuary (a) and the Vistula estuary (b).

17.3 Why coupled nitrification-denitrification was not correlated negatively particulate C:N in case of Vistula estuary?

A low C:N would indicate a high amount of N in the organic matter, which would favor nitrification via NH4+ from PON degradation. This in turn enhances denitrification coupled to nitrification in the sediment. A negative correlation of Dn with particulate C:N would thus be expected, as mentioned by the reviewer and as also found in the Öre

estuary in summer. Yet, we did not find this correlation in the Vistula estuary (see Figure 6), likely due to the overall lower C:N ratio (see lines 258-260) indicating a higher availability of N compared to the Öre estuary. In addition, the lower POC:Chl.a ratios in the BBL of the Vistula estuary (see Figure R7) suggest a high share of phytoplankton-derived POM which results in a high availability of labile organic carbon as well, the second substrate for denitrification (see line 419).

18. Line 302-303: It doesn't look so. I don't see NO3-+NO2- in BBL of estuaries differing significantly if we strictly consider standard deviation (SD) given in Table S1. On the contrary, POC and PON in BBL of Vistula estuary are much higher than that in BBL of Öre estuary (Table 2). Please rephrase these sentences.

We performed a Mann-Whitney-U-Test for significant differences between NO3-+NO2- concentrations. In summer, BBL NO3-+NO2- concentrations are significantly higher in the Öre estuary than in the Vistula estuary (U-Test, p<0.001; Figure R5 A). Also PON (U-Test, p=0.048; Figure R5 B) and POC (U-Test, p=0.04; Figure R5 C) differed significantly, although ranges are overlapping. Results of the U-Tests will be added in lines 302-303. In Table 2, there was a copy&paste mistake in the row of BBL POC and PON concentrations. We apologize for this mistake and corrected the values (see Table R2).

Figure R5: Comparison of NO3-+NO2- (A), POC (B), and PON (C) concentrations in the BBL of Öre and Vistula estuary in summer.

Table R2: Table 2 from the manuscript corrected (corrected values highlighted in yellow). Concentration of particulate organic carbon (POC) and nitrogen (PON); natural isotopic composition of POC ($\delta$13C-POC); the contribution of terrestrial and phytoplankton-derived particulate organic matter (POM) to the total POM pool measured in the river and river plume water as well as at the surface and in the bottom boundary layer (BBL) of the Öre and Vistula estuaries in spring and summer. The contribution of POM sources was estimated based on a two-component mixing model

following Jilbert et al. (2017), using end members from Goñi et al. (2003). Values are average and standard deviation of each water layer. The sample size is shown in parentheses. Site Season Water source POC ($\mu$mol L$-1$) PON ($\mu$mol L$-1$) $\delta$13C-POC (‰ Contribution terrestrial POM (%) Contribution phytoplankton POM (%)

Öre estuarya Spring River 153.6 11.2 -29.1 71 29 (1) River plume 53.7 5.1 -29.5 44 55 (1) Surface 40.2 $\pm$ 13.5 (8) 4.3 $\pm$ 1.4 (8) -25.7 $\pm$ 1.0 (8) 19 $\pm$ 16 83 $\pm$ 16 (8) BBL 36.8 $\pm$ 14.1 (10) 4.2 $\pm$ 1.5 (10) -25.0 $\pm$ 1.0 (10) 19 $\pm$ 16 81 $\pm$ 16 (10)

Summer River 67.2 5.7 -30.2 56 44 (1) River plume 46.9 $\pm$ 0.7 (3) 4.1 $\pm$ 0.7 (3) -28.7 $\pm$ 0.2 (3) 55 $\pm$ 16 45 $\pm$ 16 (3) Surface 34.1 $\pm$ 7.9 (13) 4.0 $\pm$ 0.8 (13) -26.5 $\pm$ 0.6 (13) 15 $\pm$ 11 85 $\pm$ 11 (13) BBL 135.9 $\pm$ 85.5 (9) 13.1 $\pm$ 8.4 (9) -26.1 $\pm$ 0.3 (9) 38 $\pm$ 11 62 $\pm$ 11 (9) Vistula estuaryb Spring River 164.2 16.5 -25.7 37 63 (1) River plume 61.1 $\pm$ 25.9 (8) 6.9 $\pm$ 2.5 (8) -26.5 $\pm$ 1.4 (8) 25 $\pm$ 14 75 $\pm$ 14 (8) Surface 45.6 $\pm$ 15.8 (6) 5.8 $\pm$ 2.4 (6) -24.8 $\pm$ 0.7 (6) 10 $\pm$ 16 90 $\pm$ 16 (6) BBL 25.4 $\pm$ 13.6 (18) 2.6 $\pm$ 1.3 (18) -25.6 $\pm$ 0.8 (18) 31 $\pm$ 24 69 $\pm$ 24 (18)

Summer River - - - - - River plume 103 10.2 -25.8 33 67 (1) Surface 73.6 $\pm$ 34.6 (7) 8.3 $\pm$ 3.7 (7) -25.7 $\pm$ 0.6 (7) 20 $\pm$ 10 80 $\pm$ 10 (7) BBL 46.9 $\pm$ 30.7 (11) 5.3 $\pm$ 5.5 (11) -25.4 $\pm$ 0.8 (10) 15 $\pm$ 10 85 $\pm$ 10 (9)

a Including data from Hellemann et al. (2017) b Including POC and PON concentrations from Bartl et al. (2018)

19. Line 306-309: This is true only for spring where we see high POC and PON in BBL of Öre compared to Vistula. But again on closer look, if we take SD and no. of replicates into account, POC in BBL of Öre is similar to that in Vistula and interestingly PON in BBL of Öre is higher than that in Vistula. This claim is anyway not true for summer.

Please see answer to comment no. 18. PON and POC concentrations are higher in summer than in spring in both estuaries, and higher in the Öre estuary than in the

Vistula estuary in summer (Table R2).

20. Line 328: Delete Fig.4 from the sentence as it does not show C:N.

Figure 4 shows particular C:N ratios on the y-axis of the graphs.

21. Line 330-332: Not a satisfactory explanation. Öre estuary has a sill and thus restricted exchange of estuarine water with seawater can likely cause more sedimentation within the estuary.

The reviewer is correct that there is likely more sedimentation of particulate matter within the Öre estuary. We were not aiming to contradict to this with our explanation in lines 330-332. We wanted to highlight the finding of Forsgren and Jansson (1992), that a large part of the terrestrial POM from Öre River directly sediments at the river mouth not reaching the estuary at all. Changes from line 330 ff: This was likely due to the abundant, widely dispersed estuarine phytoplankton (Fig. 2). Furthermore, Forsgren and Jansson (1992) showed that the terrestrial POM from the Öre River immediately sediments right at the river mouth and is not transported far into the Öre estuary, which may explain the small terrestrial signal in the POM from the estuarine BBL.

22. Line 333-334: C:N in Öre is higher than that in Vistula but POC:Chla in Vistula vary from 5.4 to 33.2 which is «200. How can it indicate degraded POM? Only because of C:N<12?

The reviewer may have made an error in her/his calculation. The POC:Chl.a ratios are calculated as mass ratio, i.e. $\mu$g POC L-1 (not $\mu$mol) divided by $\mu$g Chl.a L-1 (following the approach of Cifuentes et al., 1988; Savoye et al., 2003). We converted the molar concentrations of POC ($\mu$mol L-1) to mass ($\mu$g L-1) by multiplying with the molar mass of C (12.011 g mol-1). The POC:Chl.a ratios in summer are >200 in both, Vistula and Öre estuary (Table R3), indicating a low percentage of fresh chlorophyll containing biomass. For clarification, we will add this information (underlined) in section 2.2.1: lines 146-147: Particulate organic nitrogen and carbon (PON, POC) concentrations

[Figure]

($\mu$mol L-1) and... lines 145-146: The degradation state of POM was evaluated by determining the mass ratio of POC:Chl.a ($\mu$g $\mu$g-1) and molar C:N ($\mu$mol $\mu$mol-1) ratios, which increase simultaneously during degradation (Savoye et al., 2003).

Table R3: POC concentrations in $\mu$mol L-1 and $\mu$g L-1, Chl.a concentration in $\mu$g L-1, and POC:Chl.a ratios calculated from mass concentrations. Data from the BBL of Öre and Vistula estuary in spring and summer. Site Season Station POC ($\mu$mol L$-$1) POC ($\mu$g L$-$1) Chl.a ($\mu$g L$-$1) POC:Chl.a ($\mu$g $\mu$g$-$1)

Öre estuary Spring N6 37.0 444.8 7.5 60 N11 34.8 417.7 6.4 66 N11 28.1 337.0 6.4 53 NB8 33.3 399.7 3.1 131 NB8 21.6 259.2 3.1 85

Summer N6 137.1 1646.6 0.61 2684 N11 246.2 2957.5 0.60 4919 NB8 284.5 3417.1 0.55 6184 Vistula estuary

Spring VE07 14.9 178.5 1.0 174 VE07 43.7 524.5 5.7 93 VE04 38.9 467.1 2.8 164 VE06 11.2 134.8 1.0 132 VE06 28.2 338.2 1.4 247 VE18 21.7 261.1 1.9 135 VE13 11.9 143.2 1.1 131 VE13 22.9 274.8 1.5 181 VE09 12.3 147.1 1.4 108 VE09 38.8 466.3 1.3 356 VE10 10.6 127.4 1.3 96 VE10 41.3 495.7 1.7 292 VE05 11.3 135.7 1.3 104 VE05 25.6 307.2 1.3 240 VE02 56.4 677.9 2.9 232 VE49a 21.9 263.1 3.4 78 VE49a 34.7 416.4 4.9 86

Summer VE15 15.0 180.6 0.3 661 VE02 30.5 365.9 1.4 256 VE13 28.6 343.0 0.8 460 VE23 21.6 259.2 0.2 1178 VE49a 33.3 399.8 0.7 597

23. Line 336-338: Summertime POC:Chla in Ore varies from 12.6 to 140 that is <200. How can the POM be in degraded state?

Please see response to comment no. 22.

24. Line 401 and Line 397: Please show the r and p values of the correlation.

The correlation coefficient and the p value are given in line 274 and line 280.

25. Line 398: because the less-degraded POM in......This is questionable as POC:Chla

is not above 200 rather «200.

Please see response to comment no. 22.

26. Line 399: By contrast, the more degraded POM in........First of all, POC:Chla in both Öre and Vistula estuary are much lower than 200. So can we call it degraded POM? Even though we assume higher POC:Chla (>200) as indicator for highly degraded POM, POM in Vistula estuary looks more degraded compared to that in Öre estuary. Not the other way.

Please see response to comment no. 22, lines 258-260 and section 4.1.2.

27. Line 401: How significant is this correlation? What are the r and p values?

Please see line 280 and response to comment no. 15 and 16.

28. Line 402-404: These two sentences contradict each other. First sentence is questionable. I see a significant seasonal difference in the rates in both the estuaries. Please clarify the role of trophic state on these two processes.

There is no significant difference between the nitrification rates of the two estuaries (see response to comment no. 14). We agree that this short explanation is confusing and decided to remove this statement (line 401-404) from the manuscript text, as a more precise and understandable version of this is also given in the conclusions (section 5).

29. Line 407-409: Difference in denitrification........How do you know that? Where is sedimentary Corg data and $\delta$13C-Corg for both the estuaries?

As described in the results, the Vistula Estuary had more labile organic matter than the Öre Estuary based on the C:N and POC:Chl.a ratios in the BBL (see Table 2 and section 4.1.2). It is very likely that the more labile organic matter in the Vistula estuary originates from high riverine N-loads and the resulting high primary production rates. Heterotrophic denitrification uses labile organic carbon as electron donor to re-
[Figure]

duce NO3-, an increase in labile organic matter can thus increase denitrification rates (Seitzinger & Nixon 1985). Our correlation result of Dn with the LOI of the surface sediment provides evidence for this effect (see Figure 6). The organic matter content (LOI) strongly correlates with the Corg in the sediment (Figure R6). So, although we do not have sediment Corg data from the summer sampling in the Vistula and Öre estuary, we can confidently use LOI as measure of organic matter/carbon content. The $\delta$13C-Corg data give information about the contribution of terrestrial material to the total POC. $\delta$13C-Corg (-28.5 – -25 ‰ of the surface sediment in the Vistula estuary (Figure 2 in Thoms et al. 2018) are similar to $\delta$13C-POC values in the BBL (Table R2) and indicate some terrestrial contribution in the benthic POM-mixture of the Vistula estuary (see lines 326-328).

Figure R6: Correlation between LOI and Corg in the surface sediment of the Vistula estuary in spring 2016 (data kindly provided by Franziska Thoms, IOW)

30. Line 415-419: While newly produced........If higher POM availability increased denitrification rates in sediments, then why not in water? Especially in BBL?

All bottom waters were oxic (Supplement Table S1), thus the oxic-anoxic interface, where denitrification takes place, was located within the sediment. Thus denitrification in both estuaries only happened in the sediment, not in the BBL.

31. Line 421-423: what are the r and p values of the correlation?

Please see line 291.

32. Line: 425-427: Not written properly. OPD itself can get NO3- from BBL. Nitrification is significant in BBL and NO3- is not that low.

The IPT calculation (Nielsen 1992) can clearly distinguish between the NO3- source used in denitrification: the NO3- from the bottom water or the NO3- from nitrification within the oxic surface sediment (see line 225-228). The result of this calculation was that in both estuaries denitrification mainly used NO3- from nitrification within the sediment ($\geq$ 93%, line 288). In lines 425-427 we refer to the calculation result and discuss its reason.

33. Line 427: Hence, only a small......How did the authors calculate that? Is there any nitrification rate measurement in the top oxic layer of the estuarine sediments?

Please see answer to comment no. 32.

34. Line 429: This sentence contradicts the previous sentence. If the dominant NO3-source is controversial, then how can you say that <10% of NO3- from BBL was removed by denitrification in permeable sediments of Vistula estuary?

The reviewer is right, in that "controversial" was not the right word used here. We adjusted the section accordingly at line 429: In permeable sediments, the dominance of the NO3− source is highly variable due to the complexity of pore-water flow (Kessler et al., 2013; Gihring et al., 2010; Marchant et al., 2016; Rao et al., 2007). On the one hand, pore-water flow was shown to stimulate nitrification by increasing the oxic sediment volume (Huettel et al. 1998, Giehring et al. 2010, Marchant et al. 2016), and to increase the areal oxic-anoxic interface across which NO3- and NH4+ can be exchanged (Precht et al. 2004, Cook et al. 2006), thus favoring denitrification coupled to NO3- produced in the sediment (Dn; Rao et al. 2008, Marchant et al. 2016). On the other hand, pore-water flow was also shown to separate the oxic inflow from the anoxic outflow zone, limiting the exchange of NO3- and NH4+ within the sediment (Huettel et al. 1998, Cook et al. 2006, Kessler et al. 2012, 2013) and thus favoring denitrification of NO3- from the near-bottom water (Dw; Cook et al. 2006, Kessler et al. 2012, 2013, Marchant et al. 2014).

The 10% of NO3- removed from the BBL by denitrification is the result of the IPT calculation, that ~10% of denitrification was fed by NO3- from the BBL water (Dw). Please see also answer to comment no. 32.

35. Line 449: Please write it as During summer.....

Changes at line 449: During the summer cruise, permeable sediments in the Vistula estuary were not subjected to significant advective pore water flow, thus allowing the use of a diffusive incubation design.

36. Line 450-452: How is that possible? What about denitrification rate during spring?

We discuss in 4.1.3 in detail the observation, that the permeable sands of the Vistula Estuary were temporary lacking advective pore-water flow during our sampling campaign in summer 2014 (presumably due to low near bottom current velocities in summer). In the absence of advective pore-water flow, mass transport of permeable sediments is governed solely by diffusive and faunal induced fluxes, similar to cohesive sediments. We believe that the same mass transport led to the same denitrification rates, very likely due to the resulting similar transport velocity of substrates to the denitrification layer which was situated at a similar sediment depth. In spring, the permeable sands of the Vistula estuary experienced advective pore-water flow, as expected for this sediment type. Denitrification rates were however low, likely due to low availability of labile organic carbon, as well as due to problems in the incubation set-up. This issue we discuss in detail in section 4.2.3.

37. Line 457-458: Despite their......What about spring?

For the spring season, we are only able to calculate the N removal efficiency for the Vistula estuary, since no denitrification rates were detectable in the Öre estuary. The Vistula estuary removed 0.2 % of the riverine TN load via denitrification in spring (March 2016).

38. Line 461-463: These two statements are contradictory. How would the authors reconcile these statements vis-a-vis their observation?

The reviewer is right, that the statements of the cited studies are contradictory. Changes at line 461 ff: Asmala et al. (2017) estimated from a compilation of coastal denitrification rates that ∼16% of the riverine TN load entering the Baltic Sea is removed by coastal denitrification, and concluded that the Baltic Sea coastal zone is an inefficient N filter compared to the open Baltic Sea. In contrast, based on isotopic data and long-term nutrient concentrations, Voss et al. (2005a, 2011) suggested that most of the riverine N is sequestered and removed within the Baltic coastal zones. The anti-clockwise circulation pattern in the Baltic Sea, resulting in alongshore coastal jets and restricted cross-shore mixing (Radtke et al., 2012), may support coastal N retention. In this case, the coastal N filter efficiency would depend on the transport and residence time of riverine N within the Baltic coastal zone, providing time for N retention processes to recycle N until its eventual permanent removal. Accordingly, N removal efficiency alone, e.g., via denitrification rates, relative to riverine TN loads, as estimated by Asmala et al. (2017), may not be a sufficient indicator of the N filter efficiency in river dominated coastal zones.

We aimed to reconcile these statements with our observations in section 4.3. However, we see that our formulations may have not been clear enough. We will overwork section 4.3 for the revised manuscript.

39. Line 465-468: What about DNRA? That would show that how much riverine N is preserved in estuarine sediments through DNRA. It is necessary to discuss that here.

The reviewer is correct that the role of other N-transformation processes that retain N in the estuary, like DNRA, should be addressed here. Changes at line 468 ff: Accordingly, N removal efficiency alone, e.g., via denitrification rates, relative to riverine TN loads, as estimated by Asmala et al. (2017), may not be a sufficient indicator of the N filter efficiency in river dominated coastal zones. Instead, holistic approaches are needed, which also address the role of N retention processes such as nitrification or N uptake in the water column, and nitrification or DNRA in the sediment as they facilitate potential preservation of N in the coastal system.

40. 471-473: Through close......It is not necessary that only POM controlled benthic nitrification. What about benthic NH4+ efflux?

This is correct. However, we could check this relationship for a few stations where Thoms et al. (2018) measured NH4+ efflux from the sediment. These fluxes do not correlate with nitrification rates in the BBL. However, since the number of replicates is very low, we did not want to include the result in our manuscript. Nevertheless, NH4+ effluxes do supply the BBL with this nutrient which may act as important substrate source for nitrification, especially under conditions of reduced NH4+ production from POM degradation, e.g. in winter/early spring.

41. Line 482: What are the DNRA rates?

We did not measure DNRA rates in this study, please see response to comment no. 17 and response to the overall comments.

42. Line 490-492: We thus hypothesize........ How do the authors say it is a coast parallel transport? Is there any reference? The riverine flow may be perpendicular to the coast into the Baltic.

The reference is Voss et al., 2005b. Changes (underlined) from line 488-492: Furthermore, the open shape of the estuary and its unrestricted bottom topography may well enable the transport of riverine DIN and suspended estuarine POM out of the estuary and parallel to the coastal zone throughout the year (Voss et al., 2005b). We thus hypothesize that the coast-parallel transport of nutrients and estuarine POM extends the estuarine filter of the Vistula estuary to the adjacent coastal zones (Fig. 7), where microbial N retention and N removal could take place over a larger area and a longer time scale.

43. References: Holtermann et al. (2014), Risgaard-Petersen et al. (2004) and Schultz (2000) are not cited in the text. Schultz (2005) is missing in the reference list.

We thank the reviewer for checking the reference list and will correct it accordingly in the revised manuscript.

44. Table 2: The authors need to show C:N in a column here.

C:N ratios are given in section 3.1.1, lines 258-260, and in Figure 4. We think it would be too much repetition to add them in Table 2 as well.

45. Table 3: How have NH4+ surface pool and NH4+ deep pool been defined? Up to what depth you consider it as surface pool? Please mention clearly in the table caption.

The sediment NH4+ pools are defined in lines 167-169. We will add this information in the caption of Table 3.

46. Table 4: I don't see any denitrification rate in permeable sediments of Öre estuary. Was it not measured or it is not detectable? "–" symbol doesn't mean anything. Please clarify.

All sampled sandy sediments in Öre estuary were non-permeable (permeability = 0.1-0.2 × 10-12 m2, Table 3), which is discussed in detail in Hellemann et al. (2017) and mentioned in the current study in line 102, and 262-263. But, we see that in this respect, Table 4 is misleading. We will replace "-" by an appropriate abbreviation and explain it in the caption of Table 4.

47. Figure 2 & 3: The PON plots for Öre estuary are reproduced from Hellemann et al (2017). So please mention the reference clearly in the figure captions.

We thank the reviewer for pointing this out and we will add the reference to the figure caption.

48. Figure 4: This figure contradicts the data in Table 2 and Table S1. If we calculate POC:Chla from Table 2 and S1, they range from 5.4 to 140. How come Fig.4 shows such higher POC:Chla values then?

Please see the answer to comment no. 22. Although we calculated the POC:Chl.a correctly (Table R3), we found a copy&paste error in figure 4C. We apologize for this and added the corrected figure below (Figure R7). The corrected values do not change the results and discussion in the manuscript.

[Figure]

Figure R7: Corrected Figure 4: Particulate C:N ratios plotted against POC:Chl.a ratios from the surface water (A), intermediate water depths (B) and bottom boundary layer (BBL, C) of the Vistula and Öre estuaries in spring and summer. Data at intermediate water depths are water depths of 10 m and 20 m in the Vistula estuary, and 5 m and 10 m in the Öre estuary. C:N ratios: terrestrial POM (terr) > 12 according to Savoye et al. (2003); POC:Chl.a ratios: newly produced phytoplankton POM (phyt) < 200 < degraded 5 phytoplankton POM (degr) according to Cifuentes et al. (1988). Note the different scale of the POC:Chl.a ratios in panel C.

49. Figure 5: Shows vertical O2 profile of Vistula estuary sediments. But what about that of Öre estuary sediments? The authors should show that also.

The example profiles of the permeable Vistula Estuary are displayed, because they show a striking difference in O2 profile curve between spring (sigmoidal curve) and summer (parabolic curve), which we explain with presence and absence of advective pore-water flow (4.1.3). Example O2 profiles in sediments of the Öre Estuary are given in Hellemann et al. (2017) and are thus not repeated here, as the focus of Figure 5 is the presence/absence of advective pore-water flow. Nevertheless, if the reviewer feels that the manuscript benefits from showing the O2 profiles from the Öre estuary, we are will add them. Alternatively, we could add the reference for pore-water oxygen profiles of the Öre estuary in the caption of Figure 5.

50. Figure S1: The authors should point out the ports through which water sample was collected. Please point out the water above the sediments.

We updated the graphic S1. However, no water samples collected during incubation. Water circulation through the upper sediment layer was applied to mimic advective pore-water flow, and samples for N2 isotope analysis were taken at the end of incubation, after sediment and water was carefully mixed into a slurry.

1 = inflow port, 2 = potential outflow ports, 3 = actual outflow port ∼5 mm above the oxic-anoxic interface

Figure R8: Updated Figure S1: A schematic of the incubation design used to measure sediment denitrification with advective pore-water flow. Site-water spiked with 15N-NO3- tracer is pumped into the core from the top and drawn out from two sides of the oxic sediment layer (light yellow), as an approximation of the layer affected by advective flow. The outflow ports have a resolution of 5 mm, chosen according to the previously determined oxygen penetration depth.

51. Table S1: Looks a bit confusing and unexplained. River plume very much prevails within these two estuaries and occupies a depth range of up to 3m in case of Öre estuary and up to 12m in case of Vistula estuary. So when we say river plume here that actually means surface water of estuary. So, why can't the authors consider the depth from the river plume till bottom? If they do so, then I believe the so-called surface here would actually be a depth of 3m in case of Öre and 12m in case of Vistula. The authors should clear the confusion and mention terms in a logically correct way. Additionally, I believe a column for POC:Chla is necessary in this table.

We agree with the reviewer, that the given depth ranges cause confusion. The depth range of the river plumes, Öre River 3m and Vistula River 12m, which are given in section 2.1, are ranges found by previous studies (Cyberska and Krzyminski, 1988; Forsgren and Jansson, 1992). During our field campaigns, the depth range of the river plumes was ≤ 5m in both estuaries (see section 3.1.1, line 240). Within this depth range we took samples at 0m (bucket) and from the surface water with the CTD-water samplers (sampling depths: 1m-2.5m). The water samples from the remaining coastal surface (not river plume) were taken in the same depth range. Hence, water from below 5 m, belong to the mid water column. We will clarify depth ranges given in section 2.1 and in Table S1 in the revised manuscript. POC:Chl.a ratios are given in lines 255-257 and in Figure 4. We think that adding the values in Table S1 would be too repetitive. However, if the reviewer still recommends to add them, we are happy to do so.

Overall comments & suggestions: I suggest the authors to be careful about not re-peating the description of sampling methods, analysis/experiment methods and results

which are already reported by Helleman et al. (2017) and Bartl et al. (2018) for these two estuaries. For example: Do not describe the water column sampling methods, sediment sampling methods, analysis methodology, denitrification experiment method and their results in details for Öre estuary because these are already published by Helleman et al. (2017). But you can retain everything about NH4+ assimilation and nitrification in Öre estuary. Similarly, for Vistula estuary, avoid detailed description of column sampling methods, analysis methods and ammonium assimilation and nitrification experiment methods in BBL and their results because these are already published by Bartl et al. (2018). But you can retain everything about sediment sampling and analysis methods, denitrification experiment methods and their results. However, the authors can use the published data and their own generated data for the discussion since it's a comparative account study.

We thought a lot about how to structure the section 'Materials and Methods' in this manuscript, because, as the reviewer points out, it is partly repetitive to Hellemann et al. (2017) and Bartl et al. (2018). However, we came to the conclusion that it is necessary to repeat the methods shortly to ensure a comprehensive section. We do not want the reader to look up the other two publications to understand the methods we used. Instead we would like the manuscript to stand for itself. We will go through the section 2 ('Materials and Methods') again to shorten it where possible. The same holds for the presentation of the results. Some of the results reported by Hellemann et al. (2017) and Bartl et al. (2018) are given here again (always with reference), because they are needed to discuss the combined dataset in the context of our manuscript's focus.

The authors have not measured DNRA rates and have not discussed its role in transforming riverine N to NH4+ in the estuarine sediments. They have not also measured sedimentary nitrification rates which is very important. I did not see any discussion on benthic N (NO3- uptake or NH4+ release) exchange. All these could have made the discussion on benthic N cycling robust. However, the authors should use the published data (if any) on benthic nitrification, benthic DNRA and benthic N exchange and thoroughly discuss the interplay of all N cycling processes in relation to net N loss/ immobilization in these sediments in the discussion section in general and section 4.2.4 and section 4.3 in particular. I suggest the authors to relook into the classic integrated discussion on benthic N cycling in the Gulf of Bothnia by Bonaglia et al. (2017).

We agree with the reviewer that it is a drawback not to have data on DNRA and nitrification rates in the sediment. DNRA rates from coastal sediments are scarce in the Baltic Sea: muddy sediment (Jäntti et al. 2011 and 2012, Bonaglia et al. 2014, Hellemann et al. in prep), sandy sediment (Hellemann et al. in prep). These rates differ strongly between different study sites and study times. In situ nitrification rates in Baltic coastal sediment were to our knowledge so far only measured by Jäntti et al. (2011) in muddy non-permeable sediment. Further, Bonaglia et al. (2014) gives estimations of nitrification rates in muddy sediments of the Himmerfjärden (see response to comment no. 6.2). However, no data are available for sandy, permeable sediments, which comprise >50% of the area of Vistula estuary. It is thus very speculative to apply the DNRA or nitrifications rates from the literature to our study sites, but we will evaluate their potential role in the coastal benthic system of Vistula and Öre estuary. Furthermore, measurements of DNRA in coastal sediments of the Baltic Sea and a lake suggest that DNRA rates are higher at low bottom water oxygen concentrations, especially under hypoxia, and can dominate over denitrification (Bonaglia et al., 2014; Jäntti et al., 2012; McCarthy et al. 2016). During our field campaigns we did not encounter hypoxic conditions or bottom water oxygen concentrations low enough to potentially enhance DNRA over denitrification in neither of the two studied estuaries. Nevertheless, under oxic bottom water conditions coastal DNRA rates range from $1 - 487$ $\mu$mol m-2 d-1 (Bonaglia et al., 2014; Jäntti et al., 2012), and might play a significant role in coastal N retention by recycling $NO_3^-$ to bioavailable $NH_4^+$ which in turn may be further recycled within the coastal benthic system. Thus DNRA contributes to the residence time of N within the coastal zone.

The reviewer is correct, that we did not discuss the exchange of N (nitrate, ammonium) across the sediment water interface. We did not measure such fluxes ourselves, but we will check the literature to find values of such fluxes in the southern (e.g. Thoms et al., 2018 for Vistula estuary) and in the northern Baltic coastal zone and will evaluate their meaning for the manuscript's scope.

We thank the reviewer for these valuable suggestions and we will implement the role of other benthic N transformation processes and N fluxes on the coastal N filter function (retention vs. removal). However, we restrain ourselves from using rate data from other coastal sites (even though available) to calculate a benthic N-budget for our investigated estuaries as we find this too speculative. Especially, because the Baltic coastal zone is highly variable and so are the rates. Instead, we would like to adapt Figure 7 of the manuscript to show benthic N pathways in more detail, highlighting their roles as well as highlighting the gaps in our knowledge/ missing rate measurements.

In order to show the efficiency of these two estuaries as coastal filters, the authors should mention how much % of riverine N is ultimately lost in estuarine sediments through denitrification and/or anammox (if any), how much % is immobilized in sediments through DNRA and how much % is transported out of estuary to the coastal sea.

Please, see section 4.2.4, line 458, for how much % of riverine N is lost in estuarine sediments through denitrification. Unfortunately, we cannot estimate how much % N is retained in the estuarine sediments of Vistula and Öre estuary, because there are no DNRA rates available for our study sites. For the Bay of Gdansk in which the Vistula estuary is situated, model results showed that ∼46 % of the riverine TN inputs (Radtke et al., 2012) or ∼77 % of the total TN inputs (riverine, lagoon, atmospheric) are transported out of the bay. However, the resolution of the model used by Radtke et al. (2012) is too low to resolve coastal N processing, and we doubt that some of the model assumptions in Witek et al. (2003) are realistic, especially regarding the N transformation rates and the water residence time. Furthermore, no estimates

are available for the actual Vistula estuary, neither did we find results from the Öre estuary. We definitely agree with the reviewer, that it is important to discuss, how a coastal N-filter efficiency should be quantified and evaluated. We will use the valuable suggestions of the reviewer to improve our discussion in section 4.2.4 and 4.3.

Overall, I would suggest the authors to revise the manuscript by showing novelty of their study objectives, approach and findings which would make it appear as different from studies by Helleman et al. (2017) and Bartl et al. (2018).

New data that are presented in this manuscript are: 1) permeability, porosity, OPD, and pore-water $NH_4^+$ pools, and denitrification rates from the sediments of the Vistula estuary; 2) density stratification, $\delta$13C-POC, POC:Chl.a ratios, contributions of terrestrial or phytoplankton POM from the water column of the Vistula estuary; 3) pore-water $NH_4^+$ pools from the sediments of the Öre estuary; 4) density stratification, oxygen and nutrient concentrations (except bottom water, which is given in Hellemann et al., 2017), and POC:Chl.a ratios from the water column as well as nitrification and ammonium assimilation rates from the BBL of the Öre estuary. These new data were combined with published data from Bartl et al. (2018), and Hellemann et al. (2017), as the reviewer correctly states. Through this combination, we gained an extensive data set covering both, water column and sediment. This facilitated a holistic comparison of the environmental conditions in two contrasting Baltic estuaries which together with the here presented N transformation rates facilitated an approach of evaluating the coastal N filter function. We are convinced that this is a novelty compared to the studies of Bartl et al. (2018) and Hellemann et al. (2017). We are certain, that with the valuable suggestions of the reviewer, we can emphasize this novelty even more in the revised manuscript.

For your information, we summarized below, the main messages of the two published studies: Bartl et al. (2018) focused specifically on the regulation of nitrification rates in river plume and BBL of the Vistula estuary and offshore Bay of Gdansk through seasonal differences and short-term events (e.g. storm). This is a process-based

study and does not discuss the role of nitrification or ammonium assimilation in the coastal filter function. Hellemann et al. (2017) focused specifically on the N-removal process denitrification under oligotrophic conditions and emphasized the role of cohesive (non-permeable) sands, which stand in contrast to the permeable sands of the southern Baltic coast. Furthermore, the authors indeed discussed the role of denitrification for the coastal N filter function of the Öre estuary, but did not present rates of N retention processes (e.g. nitrification). The suggestions of a coastal filter function via temporary preservation of N within the Öre estuary are resumed in our manuscript and further supported by the new data presented.

Please also note the supplement to this comment:
https://www.biogeosciences-discuss.net/bg-2018-450/bg-2018-450-AC2-supplement.pdf
* * *

---

## Referee Report (RR1)

**Original comment no. 5:** *Line 163: The authors mentioned that porewater was extracted at 2 cm interval from 5 cm to 11 85 cm depth by Rhizon tubings. But Seeburg-Elverfeldt et al. (2005) says that Rhizon tubings can extract porewater with a vertical resolution of 1 cm only. Please explain.*

*Response of the authors*: Seeberg-Everfeldt et al. (2005) recommend a vertical resolution of 1 cm as highest possible resolution when sampling pore-water with rhizons. This means an interval of < 1 cm should not be applied because 90 then the pore-water catchment area of the single sampling depths would overlap and thus bias pore-water nutrient concentrations. However, an interval of > 1 cm is not problematic. At sediment depths > 5 cm, ammonium concentrations generally show a clear increasing trend in coastal Baltic sands and muds (Bonaglia et al., 2014; Lipka et al., 2018; Lenstra et al., 2018; Thoms et al., 2018) which can be well captured at a resolution of 2 cm intervals.

**Counter Comment**: *I agree with the increasing trend of porewater $NH_4^+$ in many coastal marine sediments but it is wrong to say that porewater $NH_4^+$ can be captured at 2 cm intervals. Well, let's say If you have a core of 10 cm long, you can extract porewater (by Rhizon tubings) at 0-1 cm, 1-2cm, 2-3cm, 3-4cm and so on and it would obviously represent porewater $NH_4^+$ of these 1 cm intervals. You can also extract porewater at 0-2 cm, 2-4 cm, 4-6 cm, 6-8 cm and 8-10 cm but it would not represent the porewater $NH_4^+$ of these entire 2 cm intervals rather it would represent the porewater $NH_4^+$ from 0.5-1.5 cm, 2.5-3.5 cm, 4.5-5.5 cm, 6.5-7.5cm and 8.5-9.5 cm respectively. So, it is OK to show/consider porewater $NH_4^+$ values at 1 cm, 3 cm, 5 cm, 7 cm and 9 cm in a vertical profile plot which actually means that there are some gaps in $NH_4^+$ values but nevertheless, it is OK as we get an overall increasing trend with depth.*
* * *
**Original comment no.7:** *Section 2.3.2: The authors have not given a diagram for diffusive experimental set-up.*

**Response of the authors:** Diffusive core incubations are an established and widely used incubation method for cohesive sediments e.g. Jørgensen & Sørensen 1985, Nielsen 1992, Nielsen & Glud 1996, Sundbäck et al. 2006, Hietanen & Kuparinen 2008, Jäntti et al. 2011, Bonaglia et al. 2014, Bonaglia et al. 2017. To reduce the number of figures in this paper we decided to explain the diffusive design in the text (line 192-196 of the manuscript) 160 and only show an illustration of the new advective incubation set-up, which has been designed for this study and needs detailed explanation. Nevertheless, if the reviewer feels that an illustration of the diffusive set-up is necessary, we will add one in the supplements.

**Counter Comment**: *None of the above 8 references cited by the authors has a figure of diffusive set-up. So it would be hard for the readers to visualize and understand the experiment method particularly while comparing to advective set-up. I suggest the authors to present a proper citation which actually has a figure of diffusive set-up or show a schematic diagram of the diffusive set-up.*
* * *
**Original comment no. 17.2:** *Please present few figures depicting increase in $^{15}N-N_2O$ and $^{15}N-N_2$ with time to support your conclusion on denitrification being a major N loss pathway. Similarly, if you find anammox and 340 DNRA upon re-analysis of the incubation data, then please show the proof in terms of additional figures.*

***Response of the authors:*** The presence / absence of anammox, thus its significant /non-significant contribution to total $N_2$ production and the consequential role of denitrification in N2 production were investigated by 345 concentration series (Risgaard-Petersen et al. 2003), not in time-series.

In the concentration series, D15 (= the denitrification of $^{15}N-NO3-$) has to correlate with increasing tracer concentration to fulfil basic requirements of IPT (homogeneous distribution of the tracer and nitrate limitation of the sediment, i.e. basically homogeneous uptake of the tracer, Nielsen 1992), whereas D14 (= the true denitrification) should be independent of tracer concentration, if no anammox occurs. In 350 contrast, a significant increase of D14 with increasing tracer concentration would indicate anammox, for which then separate calculations need to be applied, following Risgaard-Petersen et al. (2003). These relations were tested with regression analyses (significance level $p < 0.05$).
Below an example plot of N2 data without contribution of anammox (i.e. D14 not dependent on increasing tracer concentration: A= Öre Estuary, station N34, summer; B= Vistula Estuary, station VE05, summer), 355 as was the case in all incubations.

***Counter Comment:*** *I think it would be better if the authors show these figures in supplementary section.*
* * *
**Original comment 49. Figure 5:** *Shows vertical O2 profile of Vistula estuary sediments. But what about that of Öre estuary sediments? The authors should show that also.*

***Response of the authors:*** The example profiles of the permeable Vistula Estuary are displayed, because they show a striking difference in $O_2$ profile curve between spring (sigmoidal curve) and summer (parabolic curve), which we explain with presence and absence of advective pore-water flow (4.1.3). Example $O_2$ profiles in sediments of the Öre Estuary are given in Hellemann et al. (2017) and are thus not repeated here, as the focus of 705 Figure 5 is the presence/absence of advective pore-water flow. Nevertheless, if the reviewer feels that the manuscript benefits from showing the $O_2$ profiles from the Öre estuary, we are will add them. Alternatively, we could add the reference for pore-water oxygen profiles of the Öre estuary in the caption of Figure 5.

***Counter Comment:*** *For a comparative analysis, it would be better to reproduce porewater $O_2$ profile of Ore estuary (with proper citation) along with that of Vistula estuary*
* * *
**Original comment no. 51.** *Table S1: Looks a bit confusing and unexplained. River plume very much prevails within these two estuaries and occupies a depth range of up to 3m in case of Öre estuary and up to 12m in case 730 of Vistula estuary. So when we say river plume here that actually means surface water of estuary. So, why can't the authors consider the depth from the river plume till bottom? If they do so, then I believe the so-called surface here would actually be a depth of 3m in case of Öre and 12m in case of Vistula. The authors should clear the confusion*

*and mention terms in a logically correct way. Additionally, I believe a column for POC:Chla is necessary in this table.*

**Response of the authors:** We agree with the reviewer, that the given depth ranges cause confusion. The depth range of the river plumes, Öre River 3m and Vistula River 12m, which are given in section 2.1, are ranges found by previous studies (Cyberska and Krzyminski, 1988; Forsgren and Jansson, 1992). During our field campaigns, the depth range of the river plumes was ≤ 5m in both estuaries (see section 3.1.1, line 240). Within this depth range we took samples at 0m (bucket) and from the surface water with the CTD-water samplers (sampling depths: 1m-2.5m). The water samples from the remaining coastal surface (not river plume) were taken in the same depth range. Hence, water from below 5 m, belong to the mid water column. We will clarify depth ranges given in section 2.1 and in Table S1 in the revised manuscript.

POC:Chl.a ratios are given in lines 255-257 and in Figure 4. We think that adding the values in Table S1 would be too repetitive. However, if the reviewer still recommends to add them, we are happy to do so.

**Counter comment:** *I could not see any clarification on depth ranges in section 2.1.*
* * *
**Overall comments & suggestions:** *In order to show the efficiency of these two estuaries as coastal filters, the authors should mention how much % of riverine N is ultimately lost in estuarine sediments through denitrification and/or anammox (if any), how much % is immobilized in sediments through DNRA and how much % is transported out of estuary to the coastal sea.*

**Response of the authors:** Please, see section 4.2.4, line 458, for how much % of riverine N is lost in estuarine sediments through denitrification. Unfortunately, we cannot estimate how much % N is retained in the estuarine sediments of Vistula and Öre estuary, because there are no DNRA rates available for our study sites.

For the Bay of Gdansk in which the Vistula estuary is situated, model results showed that ~46 % of the riverine TN inputs (Radtke et al., 2012) or ~77 % of the total TN inputs (riverine, lagoon, atmospheric) 825 are transported out of the bay. However, the resolution of the model used by Radtke et al. (2012) is too low to resolve coastal N processing, and we doubt that some of the model assumptions in Witek et al. (2003) are realistic, especially regarding the N transformation rates and the water residence time. Furthermore, no estimates are available for the actual Vistula estuary, neither did we find results from the Öre estuary. We definitely agree with the reviewer, that it is important to discuss, how a coastal N-filter efficiency should be quantified and evaluated. We will use the valuable suggestions of the reviewer to improve our discussion in section 4.2.4 and 4.3.

**Counter comment:** *I could not find the section 4.2.4 in the revised manuscript. If the authors actually meant section 4.3 and 4.4, then it's OK.*
* * *
**Minor grammatical/typographical mistakes in revised version**

**Line 259:** *In coastal water column (river and river plume excluded).....*When you say coastal water column that practically means shelf waters of adjacent sea and it is out of estuary. This would be confusing for the readers. Please use an appropriate word.

**Line 334:** *.....may* **"be"** *the reason....*

**Line 436:** **.........**by increasing **"the thickness of"** oxic-anoxic interface........

**Line 456:** Replace "In the two here studied estuaries..." with "In the two estuaries studied here..."

**Line 457:** *....benthic processes* **"such as"** *nitrification,....*

---

## Author Response (AR2)

**Resubmission II of manuscript bg-2018-450**

Dear Dr. Mazumdar,

We again thank the reviewer for his/her feedback and followed the suggestions.

We hope that you find our revision satisfactory. Please, find below the response to the referee report (blue comments), a list of main changes in the manuscript, and the manuscript with tracked changes.

We would like to state at this point, that we wish to publish this study in a shared first-authorship as both scientists (Ines Bartl and Dana Hellemann) worked in equal shares on this study (please see 'author contributions' in the manuscript for more details).

On behalf of all authors, sincerely

Ines Bartl and Dana Hellemann

**Response to referee report**

**Original comment no. 5:** *Line 163: The authors mentioned that porewater was extracted at 2 cm interval from 5 cm to 11 cm depth by Rhizon tubings. But Seeburg-Elverfeldt et al. (2005) says that Rhizon tubings can extract porewater with a vertical resolution of 1 cm only. Please explain.*

*Response of the authors*: Seeberg-Everfeldt et al. (2005) recommend a vertical resolution of 1 cm as highest possible resolution when sampling pore-water with rhizons. This means an interval of < 1 cm should not be applied because then the pore-water catchment area of the single sampling depths would overlap and thus bias pore-water nutrient concentrations. However, an interval of > 1 cm is not problematic. At sediment depths > 5 cm, ammonium concentrations generally show a clear increasing trend in coastal Baltic sands and muds (Bonaglia et al., 2014; Lipka et al., 2018; Lenstra et al., 2018; Thoms et al., 2018) which can be well captured at a resolution of 2 cm intervals.

*Counter Comment*: *I agree with the increasing trend of porewater $NH_{4+}$ in many coastal marine sediments but it is wrong to say that porewater $NH_{4+}$ can be captured at 2 cm intervals. Well, let's say if you have a core of 10 cm long, you can extract porewater (by Rhizon tubings) at 0-1*

*cm, 1-2cm, 2-3cm, 3-4cm and so on and it would obviously represent porewater $NH_4$ of these 1cm intervals. You can also extract porewater at 0-2 cm, 2-4 cm, 4-6 cm, 6-8 cm and 8-10 cm but it would not represent the porewater $NH_{4+}$ of these entire 2 cm intervals rather it would represent the porewater $NH_{4+}$ from 0.5-1.5 cm, 2.5-3.5 cm, 4.5-5.5 cm, 6.5-7.5cm and 8.5-9.5 cm respectively. So, it is OK to show/consider porewater $NH_{4+}$ values at 1 cm, 3 cm, 5 cm, 7 cm and 9*

*cm in a vertical profile plot which actually means that there are some gaps in $NH_{4+}$ values but nevertheless, it is OK as we get an overall increasing trend with depth.*

**Response to counter comment:** We see that our response was not well formulated and agree with the reviewer's counter comment.
* * *
**Original comment no.7:** *Section 2.3.2: The authors have not given a diagram for diffusive experimental set-up.*

*Response of the authors*: Diffusive core incubations are an established and widely used incubation method for cohesive sediments e.g. Jørgensen & Sørensen 1985, Nielsen 1992, Nielsen & Glud 1996, Sundbäck et al. 2006, Hietanen & Kuparinen 2008, Jäntti et al. 2011, Bonaglia et al. 2014, Bonaglia et al. 2017. To reduce the number of figures in this paper we decided to explain the diffusive design in the text (line 192-196 of the manuscript) and only show an illustration of the new advective incubation set-up, which has been designed for this study and needs detailed explanation. Nevertheless, if the reviewer feels that an illustration of the diffusive set-up is necessary, we will add one in the supplements.

*Counter Comment*: *None of the above 8 references cited by the authors has a figure of diffusive set-up. So it would be hard for the readers to visualize and understand the experiment method particularly while comparing to advective set-up. I suggest the authors to present a proper citation which actually has a figure of diffusive set-up or show a schematic diagram of the diffusive set-up.*

**Response to the counter comment:** We added a schematic diagram to Fig. S2 in the supplements, so that advective and diffusive set-up are visualized and can be compared.
* * *
**Original comment no. 17.2:** *Please present few figures depicting increase in $_{15}N-N_2O$ and $_{15}N-N_2$ with time to support your conclusion on denitrification being a major N loss pathway. Similarly, if you find anammox and DNRA upon re-analysis of the incubation data, then please show the proof in terms of additional figures.*

*Response of the authors*: The presence / absence of anammox, thus its significant /non-significant contribution to total $N_2$ production and the consequential role of denitrification in N2 production were investigated by concentration series (Risgaard-Petersen et al. 2003), not in time-series. In the

concentration series, D15 (= the denitrification of $_{15}$N-NO3-) has to correlate with increasing tracer concentration to fulfil basic requirements of IPT (homogeneous distribution of the tracer and nitrate limitation of the sediment, i.e. basically homogeneous uptake of the tracer, Nielsen 1992), whereas D14 (= the true denitrification) should be independent of tracer concentration, if no anammox occurs. In contrast, a significant increase of D14 with increasing tracer concentration would indicate anammox, for which then separate calculations need to be applied, following Risgaard-Petersen et al. (2003). These relations were tested with regression analyses (significance level $p < 0.05$).

Below an example plot of N2 data without contribution of anammox (i.e. D14 not dependent on increasing tracer concentration: A= Öre Estuary, station N34, summer; B= Vistula Estuary, station VE05, summer), as was the case in all incubations.

***Counter Comment***: *I think it would be better if the authors show these figures in supplementary section.*

**Response to counter comment:** We added these figures to the supplementary section (Fig. S4).
* * *
**Original comment 49. Figure 5:** *Shows vertical O2 profile of Vistula estuary sediments. But what about that of Öre estuary sediments? The authors should show that also.*

***Response of the authors***: The example profiles of the permeable Vistula Estuary are displayed, because they show a striking difference in $O_2$ profile curve between spring (sigmoidal curve) and summer (parabolic curve), which we explain with presence and absence of advective pore-water flow (4.1.3). Example $O_2$ profiles in sediments of the Öre Estuary are given in Hellemann et al. (2017) and are thus not repeated here, as the focus of Figure 5 is the presence/absence of advective pore-water flow. Nevertheless, if the reviewer feels that the manuscript benefits from showing the $O_2$ profiles from the Öre estuary, we are will add them. Alternatively, we could add the reference for pore-water oxygen profiles of the Öre estuary in the caption of Figure 5.

***Counter Comment***: *For a comparative analysis, it would be better to reproduce porewater $O_2$ profile of Ore estuary (with proper citation) along with that of Vistula estuary.*

**Response to counter comment:** We added the pore-water oxygen profiles of the Öre estuary to Fig. 5.
* * *
**Original comment no. 51.** *Table S1: Looks a bit confusing and unexplained. River plume very much prevails within these two estuaries and occupies a depth range of up to 3m in case of Öre estuary and up to 12m in case of Vistula estuary. So when we say river plume here that actually means surface water of estuary. So, why can't the authors consider the depth from the river plume till bottom? If they do so, then I believe the so-called surface here would actually be a depth of 3m in case of Öre and 12m in case of Vistula. The authors should clear the confusion and mention terms in a logically correct way. Additionally, I believe a column for POC:Chla is necessary in this table.*

***Response of the authors***: We agree with the reviewer, that the given depth ranges cause confusion. The depth range of the river plumes, Öre River 3m and Vistula River 12m, which are given in section 2.1, are ranges found by previous studies (Cyberska and Krzyminski, 1988; Forsgren and Jansson, 1992). During our field campaigns, the depth range of the river plumes was $\leq 5m$ in both estuaries (see section 3.1.1, line 240). Within this depth range we took samples at 0m (bucket) and from the surface water with the CTD-water samplers (sampling depths: 1m-2.5m). The water samples from the remaining coastal surface (not river plume) were taken in the same depth range. Hence, water from below 5 m, belong to the mid water column. We will clarify depth ranges given in section 2.1 and in Table S1 in the revised manuscript.

POC:Chl.a ratios are given in lines 255-257 and in Figure 4. We think that adding the values in Table S1 would be too repetitive. However, if the reviewer still recommends to add them, we are happy to do so.

***Counter comment:*** *I could not see any clarification on depth ranges in section 2.1.*

**Response to counter comment:** In section 2.1, the river plumes of the study areas are described based on previous studies (lines 99 and 108), while in section 3.1.1 the extent of the river plumes during our field campaigns are presented. Information on river plume sampling was added at lines 119-120.
* * *
*Overall comments & suggestions:* *In order to show the efficiency of these two estuaries as coastal filters, the authors should mention how much % of riverine N is ultimately lost in estuarine sediments through denitrification and/or anammox (if any), how much % is immobilized in sediments through DNRA and how much % is transported out of estuary to the coastal sea.*

*Response of the authors:* Please, see section 4.2.4, line 458, for how much % of riverine N is lost in estuarine sediments through denitrification. Unfortunately, we cannot estimate how much % N is retained in the estuarine sediments of Vistula and Öre estuary, because there are no DNRA rates available for our study sites. For the Bay of Gdansk in which the Vistula estuary is situated, model results showed that ~46 % of the riverine TN inputs (Radtke et al., 2012) or ~77 % of the total TN inputs (riverine, lagoon, atmospheric) are transported out of the bay. However, the resolution of the model used by Radtke et al. (2012) is too low to resolve coastal N processing, and we doubt that some of the model assumptions in Witek et al. (2003) are realistic, especially regarding the N transformation rates and the water residence time. Furthermore, no estimates are available for the actual Vistula estuary, neither did we find results from the Öre estuary. We definitely agree with the reviewer, that it is important to discuss, how a coastal N-filter efficiency should be quantified and evaluated. We will use the valuable suggestions of the reviewer to improve our discussion in section 4.2.4 and 4.3.

*Counter comment:* *I could not find the section 4.2.4 in the revised manuscript. If the authors actually meant section 4.3 and 4.4, then it's OK.*

**Response to counter comment:** We apologize for this confusion. We changed the structure of the discussion section, so that contents of the previous section 4.2.4 are now included in 4.3 and 4.4.
* * *
**Minor grammatical/typographical mistakes in revised version**

**Line 259:** *In coastal water column (river and river plume excluded)*.....When you say coastal water column that practically means shelf waters of adjacent sea and it is out of estuary. This would be confusing for the readers. Please use an appropriate word.

→ changed to 'estuarine water column'

**Line 334: .....**may **"be"** the reason....

→ done

**Line 436: .........**by increasing **"the thickness of"** oxic-anoxic interface........

→ added 'the areal extent of the'

**Line 456:** Replace "In the two here studied estuaries..." with "In the two estuaries studied here..."

→ done

**Line 457:** ....*benthic processes* **"such as"** *nitrification,*....

→ done

**List of main changes in the manuscript**

1.  Addition of a schematic diagram for the 'diffusive' incubation set-up to Fig. S2
2.  New figure S4, depicting the requirements of the IPT/rIPT method
3.  Addition of pore-water oxygen profiles from the Öre estuary to Fig. 5
150   4. Minor structural text changes in the sections Abstract, Introduction, Materials and Methods, Results, Discussion and Conclusion, for clearer messages and a better focus

[revised manuscript text omitted]